# Atmospheric ammonia retrieval from the TANSO-FTS/GOSAT thermal infrared sounder

Yu Someya[1], Ryoichi Imasu[2], Kei Shiomi[3], Naoko Saitoh[4]

[1] Center for Global Environmental Research, National Institute for Environmental Studies, Tsukuba, Japan

[2]Atmosphere and Ocean Research Institute, the University of Tokyo, Chiba, Japan

[3]Japan Aerospace Exploration Agency, Tsukuba, Japan

[4]Center for Environmental Remote Sensing, Chiba University, Chiba, Japan

*Correspondence to*: Yu Someya (someya.yu@nies.go.jp)

**Abstract.** Hyperspectral thermal infrared sounders enable us to grasp the global behavior of minor atmospheric constituents. Ammonia, which imparts large impacts on the atmospheric environment by reacting with other species, is one of them. In this work, we present an ammonia retrieval system that we developed for the Greenhouse Gases Observing Satellite (GOSAT) and the estimates of global atmospheric ammonia column amounts that we derived from 2009 to 2014. The horizontal distributions of the seasonal ammonia column amounts represent significantly high values stemming from six

anthropogenic emission source areas and four biomass burning ones. The monthly mean time series of these sites were investigated, and their seasonality was clearly revealed. A comparison with the Infrared Atmospheric Sounding Interferometer (IASI) ammonia product showed a good agreement spatially and seasonally, though there are some differences in detail. The values from GOSAT tend to be slightly larger than those from IASI for low concentrations, especially in spring and summer. On the other hand, they are lower for particularly high concentrations during summer, such

as Eastern China and Northern India. In addition, the largest differences were observed in central Africa. These differences seem to stem from the temporal gaps in observations and the fundamental differences in the retrieval systems.

## 1 Introduction

Atmospheric ammonia plays an important role in the nitrogen cycle (Galloway et al., 2004; Behera et al., 2013). Nitrogen is taken up by animals and plants and is emitted into the atmosphere when organic matter decays or is burned.

Ammonia is one of the compounds generated as a byproduct of this process. It is well known that atmospheric ammonia reacts with acids, such as $H_2SO_4$ or $HNO_3$, and produces nitrate aerosols (e.g., Seinfeld and Pandis, 2016). These aerosols lead to widespread environmental problems, such as the formation of $PM_{2.5}$ or $PM_{10}$ that is harmful to human health, by

changing radiation budgets through the formation of cloud condensation nuclei, and by causing eutrophication through deposition into oceans or lakes.

The main emission sources of anthropogenic ammonia into the atmosphere are related to food production (e.g., ammonium-based fertilizers and domestic animals) (Behera et al., 2013). Emissions from these sources have increased along with the growth in the population after industrial evolution. The Emission Database for Global Atmospheric Research (EDGAR) provides several anthropogenic gas emission inventories, including that for ammonia. The analysis of EDGAR-HYDE data indicated that the annual global emission of ammonia into the atmosphere has increased from 8.6 TgN/yr in 1890 to 43.4 TgN/yr in 1990 (Aardenne et al., 2001). Further, data from EDGAR v4.2 were used to estimate the annual rate of increase in emissions of approximately 0.6 TgN/yr between 1970 and 2008. Biomass burning is also one of the large sources of ammonia emissions. The Global Fire Emissions Database, Version 4 (GFEDv4) (Randerson J.T. and Kasibhatla, 2017) provides information on atmospheric trace gas and aerosol emissions from fire events globally. Ammonia emissions are estimated using the burned area (Giglio et al., 2013) and an emission factor (Akagi et al., 2011). However, large uncertainties may be associated with these inventories owing to atmospheric ammonia being active and having a short lifetime. Although there are many in situ observations of ammonia, particularly in developed countries and including the Ammonia Monitoring Network (AMoN), it is quite hard to understand the global scale simply using these observational data.

Satellite remote sensing is an effective tool to get a better grasp of the global behavior of atmospheric constituents. Recent advancements in satellite sensors have allowed us to monitor minor gases, such as greenhouse gases and air pollutants. There are five recent space-borne nadir satellite TIR sounders, namely, the Atmospheric Infrared Sounder (AIRS), Tropospheric Emission Spectrometer (TES), Infrared Atmospheric Sounding Interferometer (IASI), Thermal and Near-infrared Spectrometer for Observation-Fourier Transform Spectrometer (TANSO-FTS), and Cross-track Infrared Sounder (CrIS). AIRS, IASI, TANSO-FTS, and CrIS are still being operated. Ammonia has narrow absorption bands in a spectral region of around 10 μm, and the five sounders measure thermal infrared (TIR) data in this region. The detectable phenomena should differ slightly for each sensor because of the differences in the characteristics of the sensors or because of the orbit paths of the satellites. The frontier work of these sounders for estimating atmospheric ammonia is represented by Beer et al. (2008) using TES data. The authors developed a scaling factor for the ammonia profile and converted it into an averaging kernel weighted molar fraction. Clarisse et al. (2009) revealed the highly spatially resolved global distributions of atmospheric ammonia by converting brightness temperature differences into ammonia column amounts in the atmosphere. The sensitivities of the different spectral resolutions and the thermal contrast between surface skin and surface air were investigated by Clarisse et al. (2010). Following these investigations, several other papers have presented results on the global distribution of tropospheric or column-integrated ammonia. Shephard et al. (2011) reported the spatial and seasonal variability on the basis of TES measurements, and they compared them to the GEOS-Chem model simulation. Van Damme et al. (2014, 2015) showed detailed horizontal distributions and time series derived from IASI. Shephard and Cady-Pereira (2015) investigated the detectability of ammonia and demonstrated the potential of retrieval using CrIS data. Warner et al.

(2016, 2017) detected ammonia hotspots using AIRS spectra and reported increasing concentrations from agricultural areas. However, some differences in the distributions among the products have been noted but have not been well compared to date.

In this study, atmospheric ammonia was determined using data from the TANSO-FTS on the Greenhouse Gases Observing SATellite (GOSAT). GOSAT observes both the shortwave infrared and the TIR regions simultaneously. Carbon dioxide ($CO_2$) and methane ($CH_4$) concentrations have been derived from both spectral regions (Yoshida et al., 2011, 2013; Saitoh et al., 2009, 2016) and provided as products. Therefore, the combination of NH3, CO2, and CH4 measurements within the same spatial footprint may be useful for studying linkages between the nitrogen and carbon cycles. In Sect. 2, we present the details of the satellite data and the retrieval algorithm. In Sect. 3, the horizontal and temporal distributions of ammonia column amounts derived from the GOSAT are shown. Those results are then compared with the IASI ammonia product, and the differences are discussed.

## 2 Methodology

### 2.1 Data and Models

GOSAT was launched into polar orbit on 23 January 2009. With the exception of a few time periods, data have been obtained continuously. The data cover almost the entire globe between about 85°N and 85°S. The equator-crossing time is 13:00 (local time). TANSO on GOSAT consists of FTS and Cloud and Aerosol Imager (CAI). TANSO-FTS has four bands, of which Band 4 measures the thermal infrared region between 5.5 and 14.3 µm (700–1,800 $cm^{-1}$). The spectral sampling interval is 0.2 $cm^{-1}$ and the full width at half maximum of the instrument line shape is 0.265 $cm^{-1}$. The accuracy of the spectral radiance of the TIR band of TANSO-FTS has a wavenumber dependency. Kataoka et al. (2013) reported that is it 0.5 K in unit of brightness temperature at 800 – 900 $cm^{-1}$ and 0.1 K at 980 – 1080 $cm^{-1}$. We assumed that the accuracy of spectral radiance is 0.3 K in the spectral range used in the ammonia retrieval. This is larger than those of the other hyper-spectral sounders (AIRS: ~0.15 K, IASI: ~0.2 K, CrIS: 0.05 K at 280 K). The instantaneous field of view of TANSO-FTS is 15.8 mrad circular, which corresponds to approximately a 10.5 km diameter at the Earth's surface. TANSO-FTS's scanning system adopts a two-axis pointing mechanism. The maximum pointing angle is ±35° in the cross-track direction and ±20° in the along-track direction (Kuze et al., 2009). With the exception of the target mode or sunglint mode observations over the ocean, the observation pattern was a five-point cross-track scan mode until July 2010 and a three-point cross-track scan mode since August 2010. Since the solar paddle was stopped, the operation of GOSAT was stopped on 24 May 2014. Although the operation of GOSAT restarted on 30 May 2014, the spectral resolution of TANSO-FTS had become slightly lower and the instrument line shape function was changed after the incident (Kuze et al., 2016). In this study, the L1B V160.160 products (Kuze et al., 2012) obtained from the launch to 24 May 2014 provided by the National Institute of Environmental Studies (NIES) was used. Since the magnitude of ammonia emissions to the atmosphere is much higher over land than over the oceans, and because of the computational resources required, only data obtained over land was analyzed. Cloud contaminated scenes were eliminated by CAI cloud screening (Ishida and Nakajima, 2009). The analyzed spectra are

only for daytime because CAI is observable during the daytime. In addition to cloud contamination, the dusty scenes over the desert surface were eliminated using the GOSAT slicing technique (Someya et al., 2016).

Line-By-Line Radiative Transfer Model (Clough et al., 2005) version 12.4, provided by Atmospheric Environmental Research, Inc., was used to calculate the radiative transfer and included gas absorption data obtained from the HIgh-resolution TRANsmission molecular absorption database (HITRAN) 2012 (Rothman et al., 2013). The species of $H_2O$, $CO_2$, $O_3$, $N_2O$, CO, $CH_4$, and $O_2$ were considered in the radiative transfer calculations.

The IASI L2 NH3 product version 2.1 (ANNI-NH3-v2.1) (Van Damme et al., 2017) was used for intercomparison of the results. The previous version of this product was evaluated with the data from ground-based FTS measurements from several sites (Dammers et al., 2016). This product provides ammonia column amounts for both the morning and the evening. The equator crossing time of the IASI is around 9:30 and 21:30, which are 3 h and 30 min earlier than that of GOSAT. The Interim atmospheric data from the European Centre for Medium-Range Weather Forecasts reanalysis (ERA-Interim) was used for processing this version of the products. The versions of the ERA-Interim were frequently changed, and there were discontinuities among the versions. As a result, the quality and the baseline values of the ammonia column amounts in the products could vary temporally (Van Damme et al., 2017).

## 2.2 Retrieval Algorithm

Ammonia has an antisymmetric stretch mode ($v_3$) band around $950\,\mathrm{cm}^{-1}$. Figure 1 illustrates the observed GOSAT spectrum over Northern India, its residual between the observed and calculated spectra, and ammonia transmittance. The calculated spectrum was constructed using the a priori temperature and water vapor profiles. In this study, the band between 960 and $969\,\mathrm{cm}^{-1}$ was utilized for the retrievals because the band near $930\,\mathrm{cm}^{-1}$ overlapped with those of the other species. The ammonia retrieval algorithm is based on the nonlinear optimal estimation method (Rodgers, 2000) with the Gauss–Newton iteration represented as the following equation:

$$x_{i+1} = x_a + \left(S_a^{-1} + K_i^T S_\epsilon K_i\right)^{-1} K_i^T S_\epsilon^{-1} [y - F(x_i) + K_i(x_i - x_a)], \tag{1}$$

where $x$ is the state vector, $S_a$ and $S_\epsilon$ are the covariance matrices for a priori and measurements, respectively, $K$ is the Jacobian matrix, $y$ is the measurement vector, $F(x)$ is the forward model vector, and the subscripts $i$ and $a$ denote the number of iterations and an a priori value, respectively. In this study, the shape of the ammonia mixing ratio profile follows that of the Air Force Geophysics Laboratory (AFGL) atmospheric constituent profiles (Anderson et al., 1986) from which the scaling factor for the profile was retrieved. The standard deviations of the a priori scaling factor and the measured spectra for ammonia retrievals were assumed to be 20 and 0.3 K, respectively. The ammonia profile of the AFGL profiles and the column averaging kernel for the a priori value is shown in Figure 2. The 2D plot of the averaging kernel matrix is also shown in Fig. S1. If the retrieved scaling factor is negative, the column amount will be recognized as zero. The ammonia profile was scaled using the retrieved scaling factor and converted to a column-integrated amount.

Since temperature and water vapor profiles significantly affect the ammonia retrieval results, they should be considered appropriately. In this study, they are retrieved from the measured spectra in advance of the ammonia retrievals. The retrievals are performed sequentially according to Table 1. The a priori information of temperature and water vapor is estimated from linear temporal and spatial interpolations of the Grid Point Value (GPV) of Global Spectral Model (GSM) products provided by the Japan Meteorological Agency. The GPV (GSM) data are provided for four times per day and have a spatial resolution of 21 layers vertically and $0.5° \times 0.5°$ grids horizontally. The profiles of $CO_2$ and $CH_4$ were obtained from the NIES transport model (Saeki et al., 2013). In each retrieval step, the surface temperature was retrieved simultaneously. Surface emissivity was inferred from the Advanced Spaceborne Thermal Emission and Reflection Radiometer Spectral Library (Baldridge et al., 2009) for the land cover types obtained from the MODIS land cover products for each scan (Friedl et al., 2010) based on the International Geosphere-Biosphere Programme land cover classification.

## 3 Results

Figure 3 contains seasonal maps of the horizontal distributions of the ammonia column amounts obtained from GOSAT within $2.5° \times 2.5°$ grids for each season. Considerably high values were observed above several areas such as India, eastern China, and central Africa. Figure 4 shows the averaged errors within each grid. When the estimated amounts were negative, the values were excluded. The errors were approximately $0.5 \times 10^{16}$ molec/cm$^2$. The high concentration areas are highlighted in Fig. 5 and listed in Table 2. Figure 6 depicts the distributions of anthropogenic ammonia emissions obtained from version 2 of the Emission Database for Global Atmospheric Research–Hemispheric Transport of Air Pollution (EDGAR-HTAP; Janssens-Maenhout et al., 2012), and from GFED4.1s. Similar to Fig.3, the ammonia column amounts over land obtained from IASI are shown in Fig. 7. The time period of these observations was the same as that for the GOSAT data and was from 23 April 2009 to 14 December 2014. The data were filtered to obtain data collected only during the day and in the absence of clouds. The differences between Fig. 3 and Fig. 7 are shown in Fig. 8. The time series of the monthly averaged ammonia column amounts obtained from GOSAT and IASI for six agricultural emission areas (central U.S., Europe, central Asia, India, Southeast Asia, and eastern China) and four biomass burning areas (South America, central Africa, South Africa, and western Russia) are outlined in Table 2 and are shown in Fig. 9 and 10, respectively.

## 3.1 Ammonia hotspots and their sources

The horizontal distribution of hotspots and their seasonal variations are clearly seen in Fig. 3. The values were high in the spring and summer seasons across most of the areas. The distribution of these hotspots is quite similar to those previously reported using TES (Shephard et al., 2011), IASI (Van Damme et al., 2015), and AIRS (Warner et al., 2016). The largest values were observed in northern India. In this area, the values were high throughout the year and were more than 1.2 molec/cm$^2$ during JJA. According to these figures, anthropogenic emissions predominate in India. This area is known to be the strongest anthropogenic ammonia emission source in the world, stemming from agricultural activities, such as

livestock and fertilizer applications. The high concentrations also observed in eastern China, central U.S., and Europe during MAM and JJA are similarly due to agricultural emissions. China, in particular, uses the largest amount of nitrogen fertilizers in the world (Lu and Tian, 2017), as corroborated by the correspondingly high concentrations of ammonia detected. In spite of the smaller emissions indicated by the EDGAR data, the results from GOSAT show high values in central Asia. In this area, the application of nitrogen fertilizers for cotton production has increased, in addition to the number of domesticated sheep (Huang et al., 2012). High concentrations for this area were also noted for in situ observations (Li et al., 2012) and those reported by Van Damme et al. (2014) using the IASI data and by Warner et al. (2016) using the AIRS data. Hence, the high values obtained from the GOSAT probably reflect these emissions. On the other hand, emissions from biomass burning dominate in South America, central Africa, South Africa, and western Russia as seen in GFED. Peak values were noted in central Africa during MAM and in South America and South Africa during SON. The high values noted in JJA for western Russia were a result of large-scale biomass burning that occurred during the disastrous heatwave in 2010. With the exception of central Africa, the temporal variations in these areas agreed well with those obtained from the MODIS fire count product (not shown). In central Africa, temporal variations are not consistent with the MODIS product, as is discussed in Sect. 4. Both anthropogenic and biomass burning emissions are seen in Southeast Asia inventories. The peak was observed in MAM, although only a few observations were available for JJA owing to the high cloud occurrences.

### 3.2 Comparison of the GOSAT and IASI L2 product

Results from GOSAT are compared with the IASI ammonia product. The horizontal distributions and seasonal variations observed for GOSAT and IASI data in Fig. 3 and Fig. 7 show similar patterns. Nonetheless, some differences were noted. As a whole, the values from GOSAT tend to be larger than those from IASI in Fig. 8. The differences were especially large in several areas. The values from GOSAT were much higher than those from IASI in such as western Africa and southern India during JJA. On the other hand, they were smaller in the coastal area of central Africa during DJF, in northern India during JJA, and in eastern China during JJA. These differences are discussed in the next section.

In Fig. 9, the peak term was short (i.e., one or two months) in India, Southeast Asia, and eastern China. On the other hand, in central U.S., Europe, and central Asia, the peak terms were relatively long, about three or four months. The temporal variation patterns in the values obtained from GOSAT and IASI were similar, although some differences in the magnitude were noted as per the results shown in Fig. 8. In central U.S., the values from GOSAT and IASI were in good agreement. With the exception of 2010, GOSAT estimated higher values in the peak season over Europe. A similar trend was noted for both central Asia and Southeast Asia. The differences were especially large for central Asia. Although the peak values were not so different for India and eastern China, the GOSAT values in the bottoms were higher. In Fig. 10, the variations of peak value between each year over South America were larger than those over the other areas. In contrast, the peak values in South Africa were very similar across the years. This should be reflected in the difference in scale of biomass burning among the years. For example, only the value of August 2010 for western Russia was quite large, which is attributed to the large-scale wildfires that occurred during this period. The consistency between the results from GOSAT and IASI in

these areas is higher than that observed in Fig. 9. However, the largest difference was observed over central Africa. Although IASI shows peaks around March, the peak for GOSAT appeared in summer and the values from March were much smaller than those from IASI. One of the causes for these high summer values from GOSAT is probably dust aerosol contamination, as discussed in Sect. 4.

## 4 Discussion

The comparison between GOSAT and IASI product results showed a similarity in the spatial and temporal distributions. However, some differences larger than the level of estimated errors shown in Fig. 4 were also found as noted in the previous section. The possible main causes generating differences are (1) local time of observation, (2) signal-to-noise ratio (spectral resolution and noise level), (3) scan geometry, (4) assumed ammonia profile shape, and (5) cloud/aerosol screening. In the previous section, the differences between the values from GOSAT and IASI were relatively large above agricultural areas (see Fig. 8 and Fig. 9).

The temporal gap of (1) is probably to incur one of the most significant differences in the measured values. The sensitivity of the TIR measurements changes diurnally because of the diurnal thermal contrast between surface skin and air. In many cases, the surface temperature at the GOSAT measurement time is higher than that at the IASI measurement time during the daytime. As a result, the thermal contrast was larger at the GOSAT measurement. Clarisse et al. (2010) compared ammonia retrievals using both IASI and TES, which have observation times close to that of GOSAT and a higher spectral resolution than that of IASI. They reported that the thermal contrast was larger, the spectral resolution was higher, and the sensitivity to the lower level was higher. Therefore, GOSAT should be more sensitive to the lower atmospheric levels where, in many cases, the concentration of ammonia is generally higher than that in the high levels as compared to IASI. This can explain the differences are smaller or in opposite in the summer. The fact that the differences are smaller at high latitudes also supports the large contribution of thermal contrast differences. In addition, the anthropogenic ammonia emissions should be temporally variable. There is about 3.5 h gap between GOSAT and IASI observations. Several studies have investigated the diurnal variations in atmospheric ammonia concentrations near the surface. According to these studies, the variations are especially large over agricultural areas during the summer season, with patterns differing for each location (e.g., Erisman et al., 2001, Meng et al., 2011, Sharma et al., 2014, Wang et al., 2015). These reports estimated that the variations fluctuate along with the temporal variations in human activities, such as fertilization and traffic. However, the differences caused by diurnal cycles of ammonia should be larger during summer. Therefore, the contribution from this seems to be smaller than that from the thermal contrast.

Cause (2) is related to the detection limits of the instruments used. The spectral resolution of GOSAT ($0.265$ cm$^{-1}$) is higher than that of IASI ($0.5$ cm$^{-1}$), AIRS ($0.5$ cm$^{-1}$), and CrIS ($0.625$ cm$^{-1}$). The finer spectral resolution provides better isolation of ammonia signals in the spectra. Moreover, high spectral resolution can reduce the contaminations from interfering species, such as water vapor, by selecting narrower fitting windows that exclude spectral features of other

gases. If the AFGL ammonia profile is assumed, the maximum signal of ammonia is approximately 0.04 or 0.05 K for a spectral resolution of 0.5 or 0.6 cm$^{-1}$, which corresponds to AIRS, IASI, and CrIS. On the other hand, it is approximately 0.1 K for a resolution of 0.2 cm$^{-1}$, which corresponds to GOSAT (see Fig. S3). This indicates that the isolation of the ammonia signal in the GOSAT spectra is approximately twice as good as those in the other sounders if they have the same noise levels.

In Sect. 2.1, we assumed a spectral accuracy of 0.3 K. If the relation between the signal and the ammonia concentration is linear, then the ammonia column amount corresponding to 0.3 K of the ammonia spectral signal at the strongest channel is approximately $1.4 \times 10^{16}$ molec/cm$^2$. This value is considerably higher than the minimum values seen in the previous figures. Based on these figures, the lower limit of GOSAT retrieval seems to be approximately $0.2 \times 10^{16}$ – $0.4 \times 10^{16}$ molec/cm$^2$. This corresponds to an ammonia spectral signal lower than 0.1 K in the spectra. Although our assumption of spectral noise is

based on Kataoka et al. (2013), the reported values not only include random noise but also spectral biases. Therefore, our results imply that the random noise of the GOSAT spectral region used in the ammonia retrieval is approximately 0.1 K.

Over smaller and more scattered agricultural areas as compared to hotspots, such as those noted for Europe, (3) likely also contributes to the differences noted in Fig. 9.

Cause (4) possibly affects the column integration of ammonia amounts. In this study, the ammonia profile shape is

assumed to be that of the AFGL profiles. However, the vertical gradients of the concentrations can be considerably larger in source regions and considerably smaller in clear air. On the other hand, IASI retrieval uses fitting function profiles based on the Goddard Earth Observing System chemical transport model and the parameters characterizing the shape of ammonia profile are retrieved (Whitburn et al., 2016).

In Fig. 8 and 10, the largest differences were observed over central Africa. The time series from GOSAT has two peaks

around March and July. The peaks in March are consistent with those observed by Whitburn et al. (2015), who reported that these peaks, observed from the IASI product, were related to biomass burning at the end of the boreal winter in this area. In another study, Hickman et al. (2018) reported that the high concentrations of NH$_3$ in March and April observed by IASI are related to the soil moisture. In this study, the dusty scenes were eliminated by the CO$_2$ slicing technique that uses the TIR region in addition to CAI cloud screening as mentioned Sect. 2 because it is very hard to detect dust aerosols over desert

surfaces using only a visible imager. If the CO$_2$ slicing is not used for the screening, large apparent ammonia column amounts are calculated over the Saharan desert in JJA, where no emission sources are present, as a result of the frequent dust storms that occur in this region. However, the dusty scenes are not completely eliminated by this technique (Someya et al., 2019). Figure S2 shows an example of the observed spectra from which a high ammonia value was retrieved in the Sahara. The spectra are V–shaped as a result of decreasing radiances with wavenumbers between 800 cm$^{-1}$ and 1000 cm$^{-1}$ and

increasing radiances with wavenumbers between 1060 cm$^{-1}$ and 1240 cm$^{-1}$. This shape is a characteristic of dust contamination due to the spectral dependence of the refractive index. Therefore, a part of the high values in northwestern Africa during JJA were likely caused by dust contamination. However, further investigation is needed because the composition of dust and its impact are quite complicated. Moreover, other error sources such as surface emissivity and humidity also exist. On the other hand, the high values obtained by GOSAT over central Africa in the boreal spring and

summer may not only result from these contaminations. The seasonal variations observed by AIRS and reported by Warner et al. (2016) showed similarly high concentrations in central Africa during MAM and JJA. This area is characterized by a number of distributed surface types whose temporal patterns of ammonia concentrations are different for each. Figure 11 shows the monthly variations in ammonia column amounts over six surface types, derived from GOSAT. Peaks over forests or woody savannas were observed in February or March, similar to the IASI observations. In contrast, peaks were noted in the boreal summer over open shrublands, grasslands, and croplands. Adon et al. (2010) reported that the peaks in atmospheric ammonia concentrations are more commonly detected during the wet season (boreal summer) over dry savannas, whereas those detected during the dry season (boreal winter) were found over wet savannas and forests. Herein, grasslands are included in the dry savanna categorization as per Adon et al. (2010), and the observed patterns in Fig. 11 are consistent with this report. Therefore, the peaks in the boreal summer observed by GOSAT might be a result of the high concentrations over surfaces such as grasslands and open shrublands. Further studies are required to investigate the source, sink, and concentration patterns and to understand the dynamics of ammonia in this area.

The discussions above are only speculations. We must evaluate the observed differences considering other independent observations.

**5 Summary**

Ammonia column amounts were retrieved using the thermal infrared spectra obtained from GOSAT for a period of approximately five years. The optimal estimation method, which iteratively minimizes the difference between the calculated and the observed spectra, was used for analysis. Temperature and water vapor profiles were also retrieved sequentially in order to reduce retrieval errors. The gridded horizontal distributions of atmospheric ammonia column amount for each season (DJF, MAM, JJA, and SON) were shown. Significantly large ammonia column amounts were noted in central U.S., South America, Europe, central Africa, South Africa, western Russia, central Asia, India, Southeast Asia, and eastern China. These distributions are similar to those previously reported using the other sounders. The areas categorized as anthropogenic and biomass burning emission source regions, based on inventories and characteristics, were investigated. The hotspots noted in the data were consistent with those from the inventories. However, high values were detected in central Asia in spite of the lower inventoried emissions. These high values likely resulted from the application of fertilizers for cotton production, as reported by in situ observations. The regional time series of the concentrations in those areas clearly detect the high and low seasons. A comparison with the IASI ammonia product showed a good agreement horizontally and temporally, although there were some differences in the details. The values obtained from the GOSAT data tend to be large relative to the background level of concentrations in the spring and summer. Differences are especially large in areas and during seasons with heavy fertilizer application. The main cause of these differences seems to be temporal gaps in the observations, especially caused by the thermal contrast between surface skin and air. In the biomass burning emission areas, the consistency is relatively high. The largest differences were noted in central Africa. These differences may stem from the

contaminations such as dust aerosols and the variation in seasonal patterns over various surface types. However, further investigation of these differences is needed because the contaminations are complicated in this area.

Although the results were evaluated by comparing them with the IASI product, they should be validated with the other measurements. The ground-based measurement is appropriate for validation. The IASI and CrIS ammonia products were validated with the ground-based Fourier Transform Infrared Spectroscopy (FTIR) measurement at the Network for the Detection for Stratospheric Change (NDACC) sites (Dammers et al., 2016; Dammers et al., 2017). Unfortunately, very few coincident measurements exist between GOSAT and NDACC FTIR with the match-up criteria shown in the papers (90 min and 50 km) because of the sparse scan geometry of GOSAT. Therefore, we must validate our results as a part of future work. Moreover, other satellite ammonia products exist from AIRS and CrIS. Inter-comparisons with these products are also needed. This should lead to a higher reliability of the satellite products and a deeper understanding of ammonia behavior.

In a recent paper, it was reported that the use of nitrogen fertilizers has been increasing (Lu and Tian, 2017) and that this trend will continue with the growth of the human population. Hence, it is urgent to continue monitoring global atmospheric ammonia in order to understand its influence on the environment. The same analysis will be applicable to data from GOSAT-2, which was launched in October 2018. This will lead to the construction of a long-term ammonia database. In addition, GOSAT-2 will be able to observe $CO_2$, $CH_4$, and CO simultaneously with ammonia. This combination of observed products is useful for better understanding of the role of specific phenomena, such as biomass burning and agricultural activities.

**Acknowledgments**

This work was supported by JSPS KAKENHI Grant Number 17K17670 and the program of JAXA (JX-PSPC-455165). The computational resources were partly provided by NIES-RCF2. The IASI-$NH_3$ products were accessed in August 2019 from http://cds-espri.ipsl.fr/etherTypo/index.php?id=1700&L=1 and https://iasi.aeris-data.fr/nh3/. IASI is a joint mission of EUMETSAT and the Centre National d'Etudes Spatiales (CNES, France). The authors acknowledge the AERIS data infrastructure for providing access to the IASI data in this study and ULB-LATMOS for the development of the retrieval algorithms.

**Author contributions**

YS contributed to the development of the retrieval system, analysis of the satellite data, preparing the manuscript, and funding acquisition. RI contributed to funding acquisition, and literature review. KS and NS contributed to data provision.

**Conflicts of interest**

The authors have no conflicts of interest, financial or otherwise, related to this study.

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

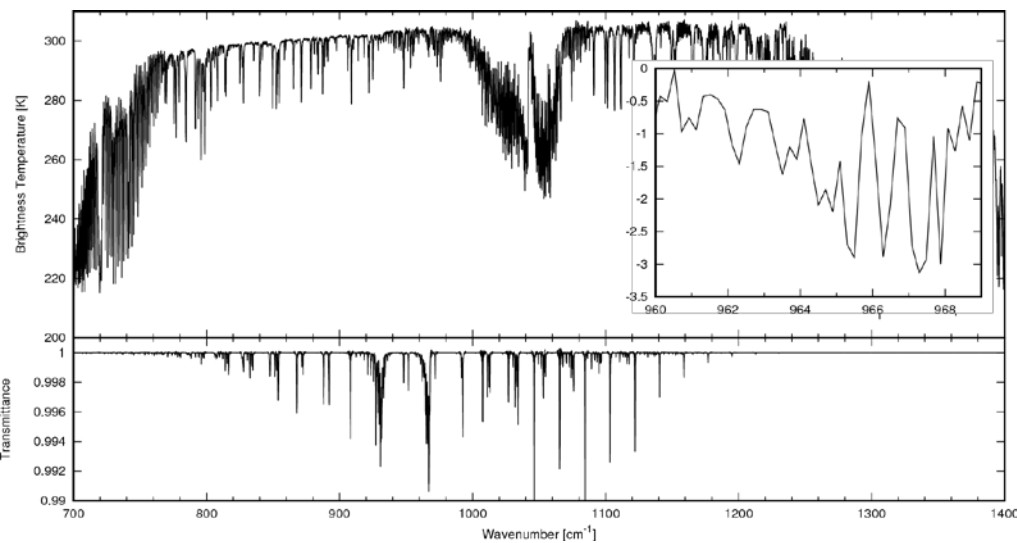

**Figure 1:** Observed spectra by the GOSAT at (75.85°E, 29.80°N) on 15 July 2013 (top), the residuals between the observed spectra and the one calculated without NH₃ (right inside), and NH₃ transmittance using the AFGL profile (bottom).

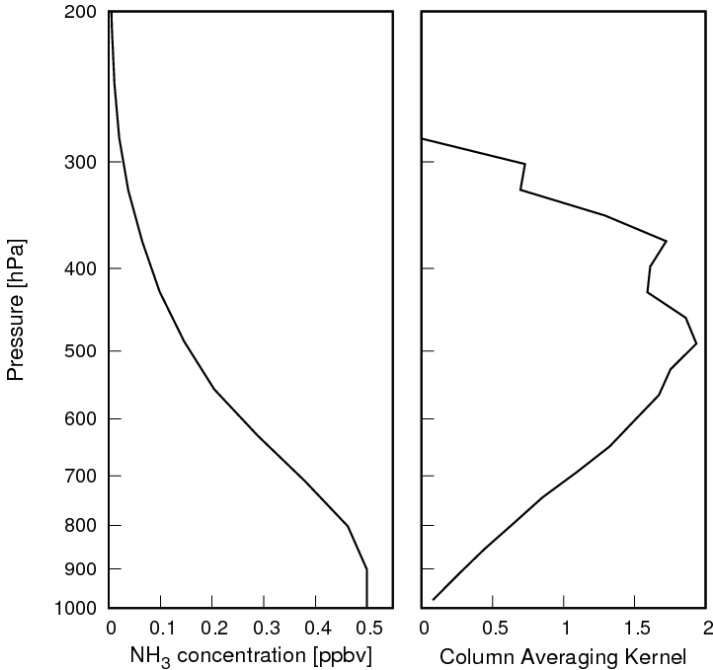

**Figure 2:** Ammonia concentration of the AFGL profiles (left) and the column averaging kernel for the a priori ammonia profile using mid-latitude summer atmosphere (right).

**Table 1:** Wavenumber range and a priori information used in the retrieval steps.

| Retrieved parameter | Wavenumber range ($cm^{-1}$) | A priori |
|---|---|---|
| Temperature | 720–780 | GPV (GSM) |
| Water vapor | 1,205–1,245 | GPV (GSM) |
| Ammonia profile | 960–969 | AFGL profile |

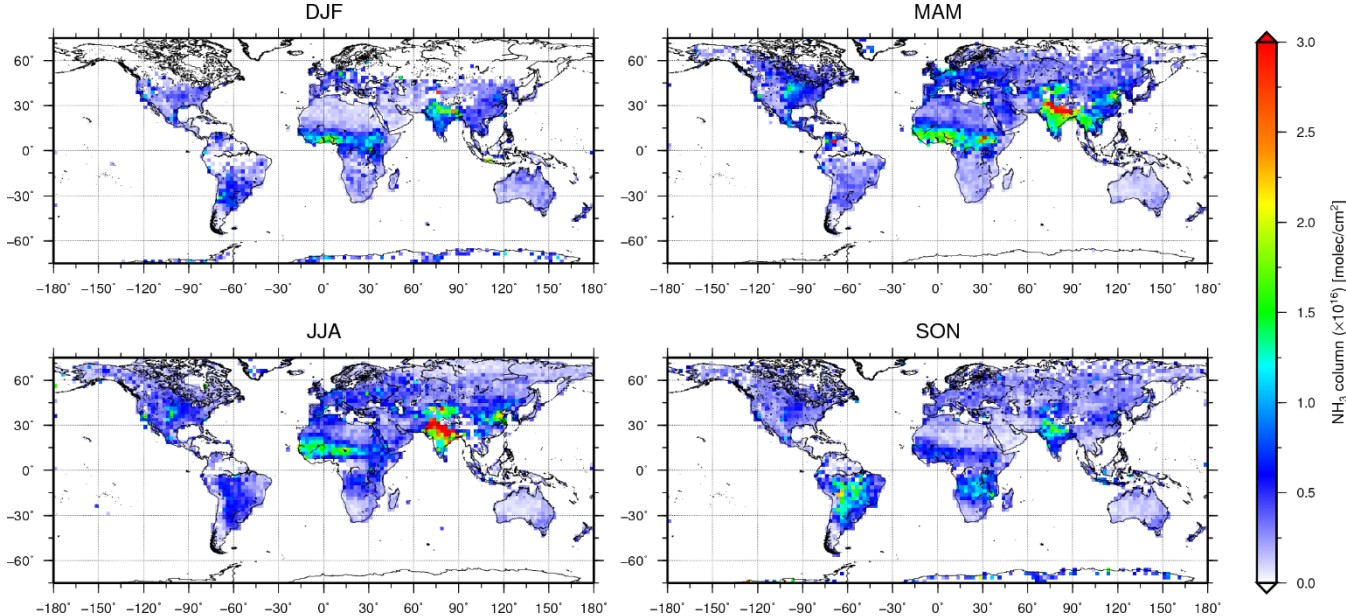

**Figure 3:** Horizontal distributions of ammonia column amounts for each season: December through February (DJF); March through May (MAM); June through August (JJA); and September through November (SON); as retrieved from the GOSAT over 2.5° × 2.5° grids averaged from April 2009 to May 2014.

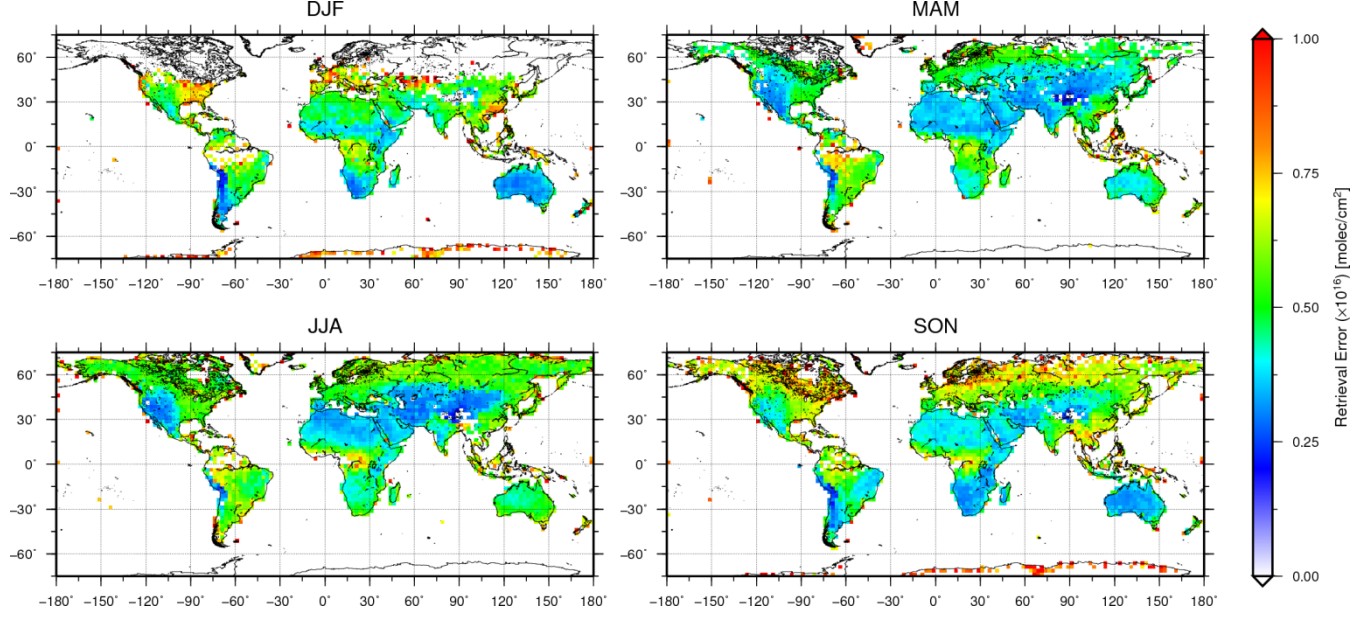

**Figure 4** Same as Figure 3 but for averaged errors.

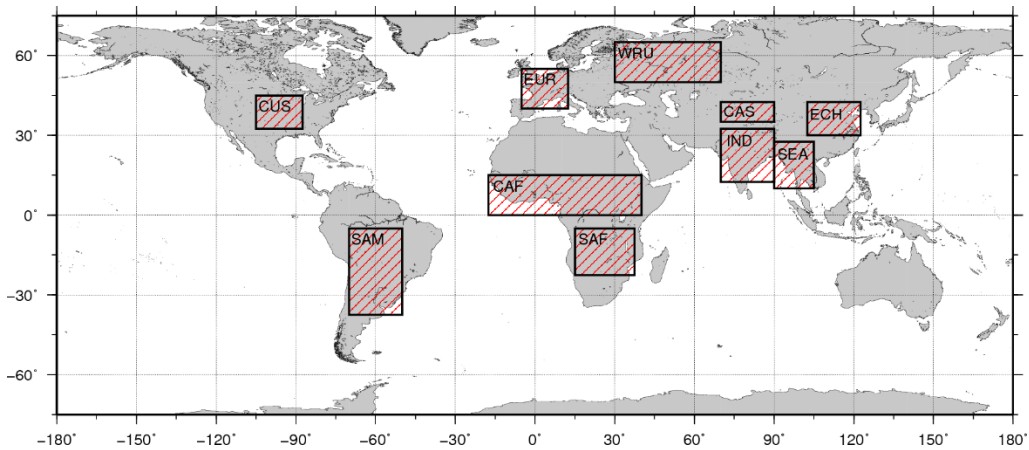

**Figure 5:** Distribution of areas with high ammonia concentrations obtained from the GOSAT.

**Table 2:** List of areas indicated in Figure 5 and their locations.

| Area | Location |
| --- | --- |
| Central U.S. (CUS) | (32.5°N–45°N, 105°W–87.5°W) |
| South America (SAM) | (37.5°S–5°S, 70°W–50°W) |
| Europe (EUR) | (40°N–55°N, 5°W–12.5°E) |
| Central Africa (CAF) | (0°–15°N, 17.5°W–40°E) |
| South Africa (SAF) | (22.5°S–5°S, 15°E–37.5°E) |
| Western Russia (WRU) | (50°N–65°N, 30°E–70°E) |
| Central Asia (CAS) | (35°N–42.5°N, 70°E–90°E) |
| India (IND) | (12.5°N–32.5°N, 70°E–90°E) |
| Southeast Asia (SEA) | (10°N–27.5°N, 90°E–105°E) |
| Eastern China (ECH) | (30°N–42.5°N, 102.5°E–122.5°E) |

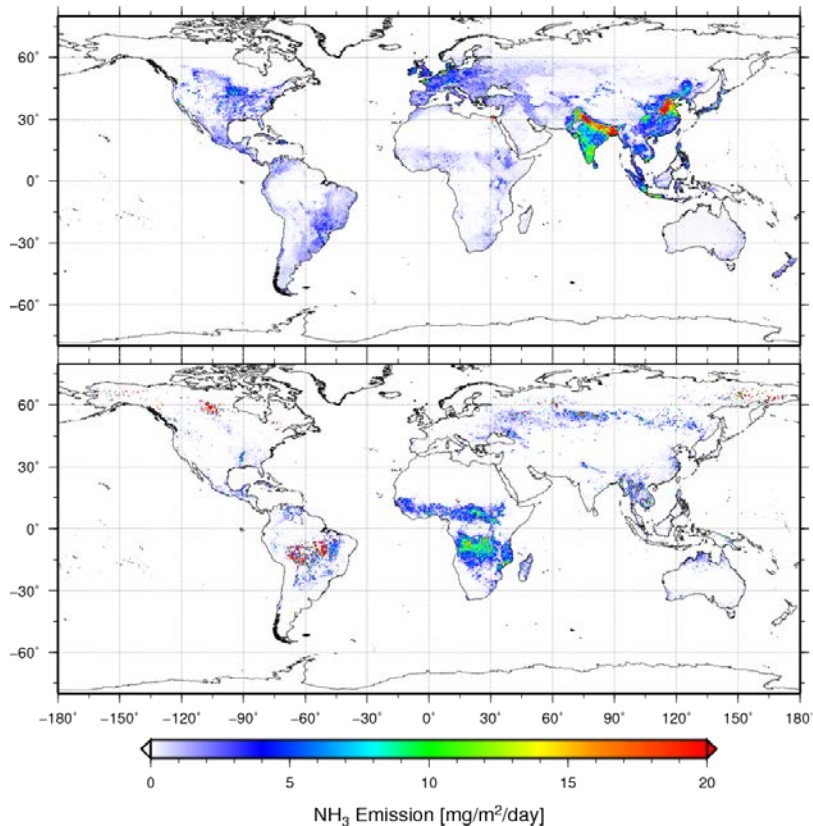

**Figure 6:** Anthropogenic ammonia emissions in 2010 obtained from EDGAR-HTAP v2 (top) and the biomass burning emissions from GFED4.1s (bottom).

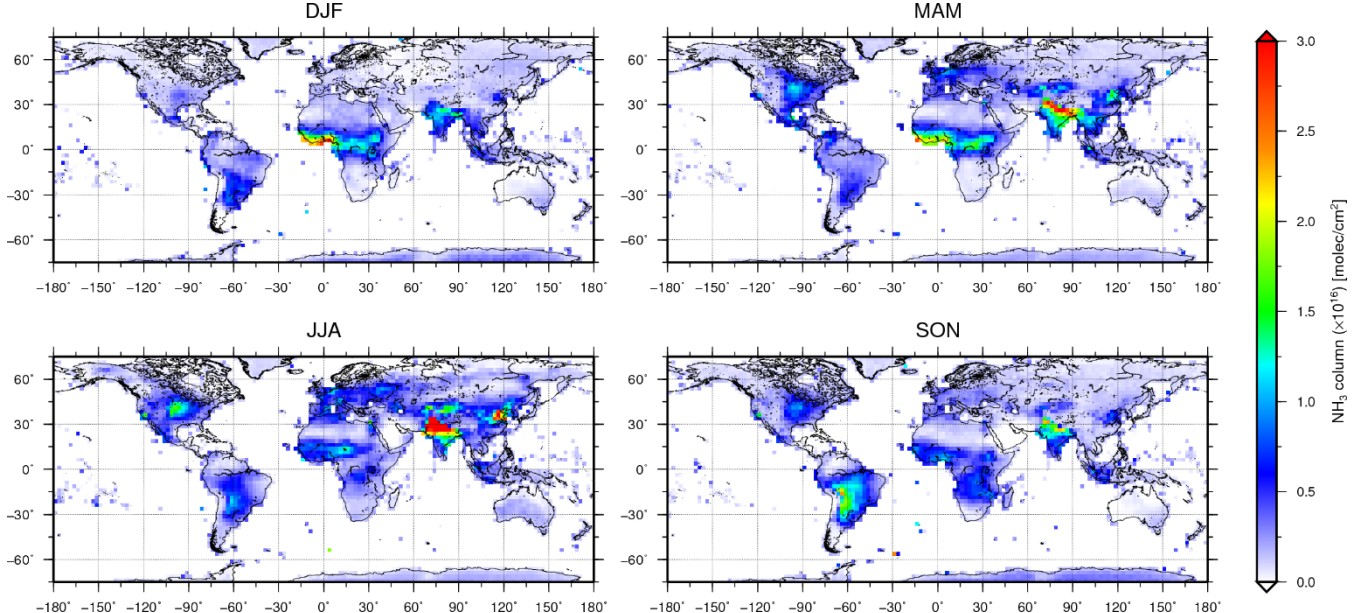

**Figure 7:** Same as Figure 3, but for the IASI measurements.

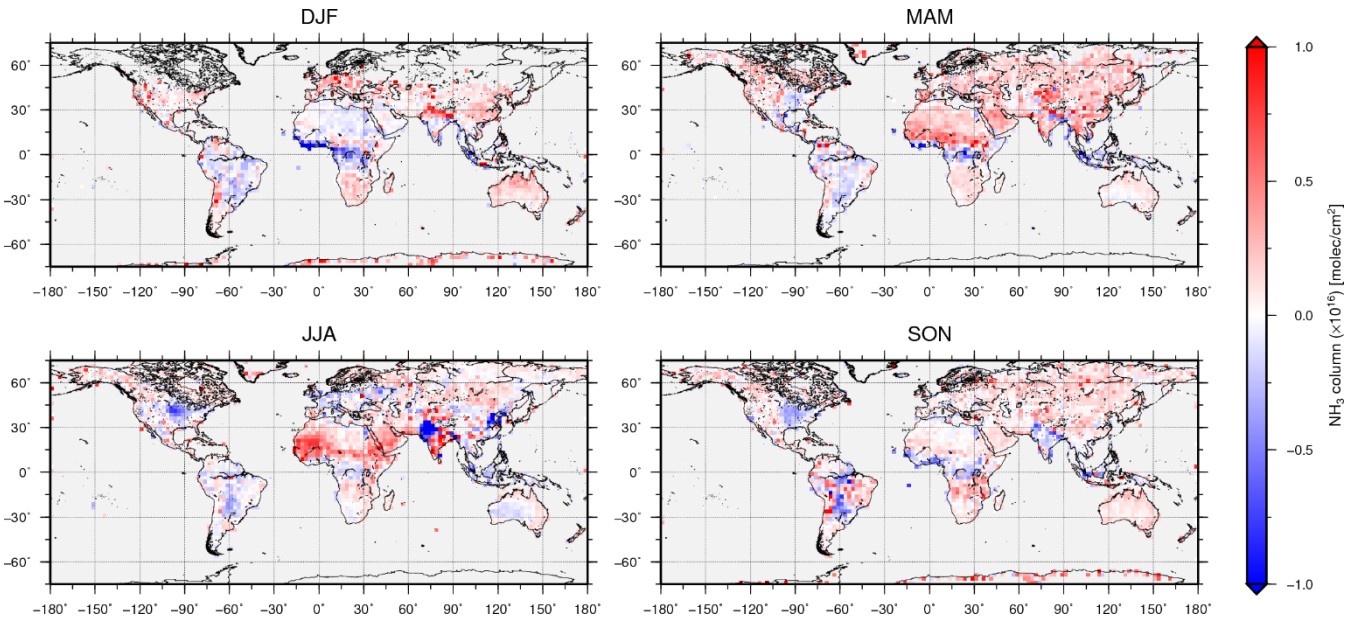

**Figure 8:** Same as Figure 3, but for the difference between the GOSAT and the IASI ammonia column amounts. The red color indicates that the GOSAT data are higher than that from the IASI.

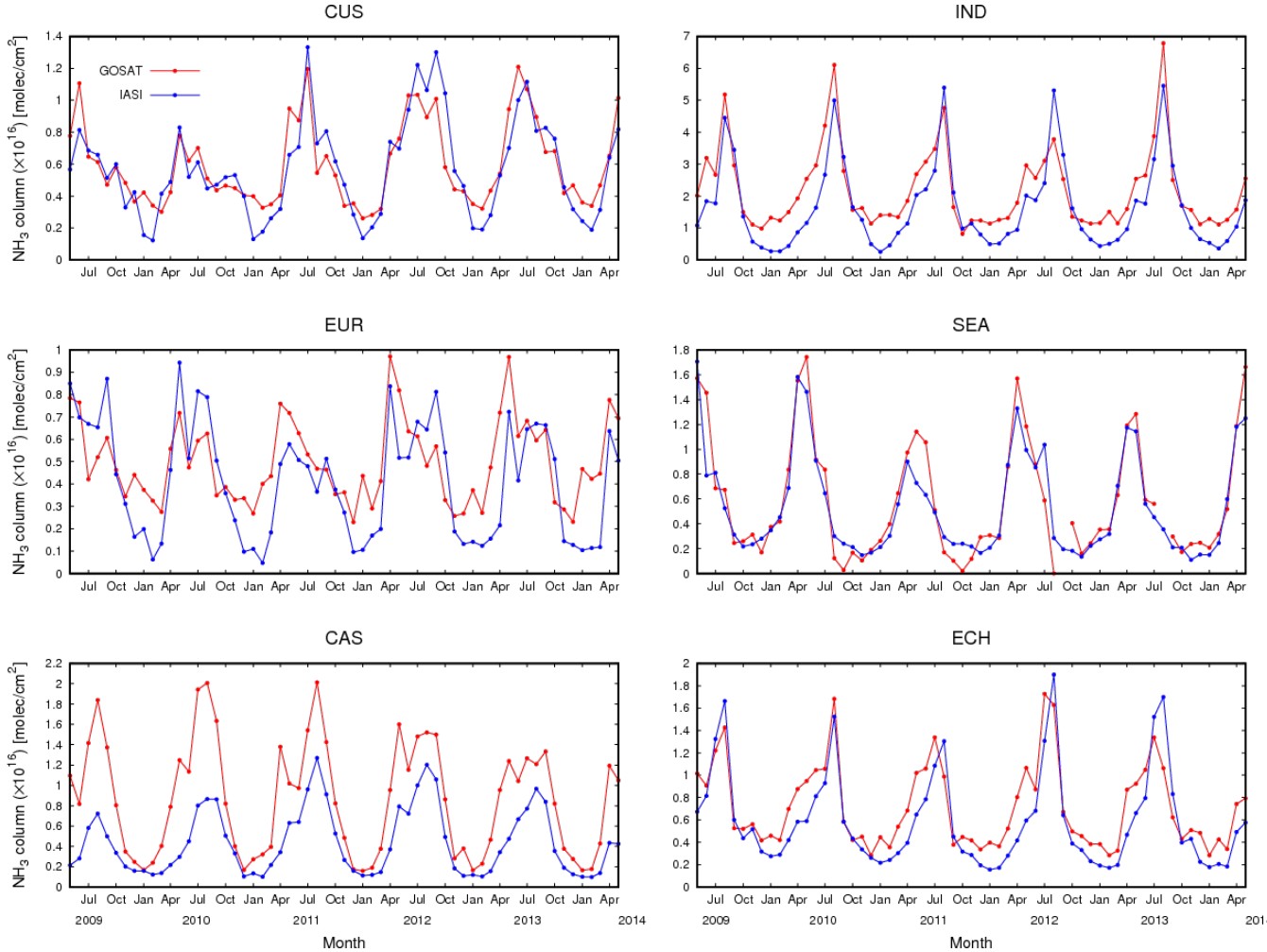

**Figure 9:** Time series of the monthly averaged ammonia column amounts from the GOSAT (red) and IASI (blue) from May 2009 to May 2014 in the agricultural emission areas noted in Table 2.

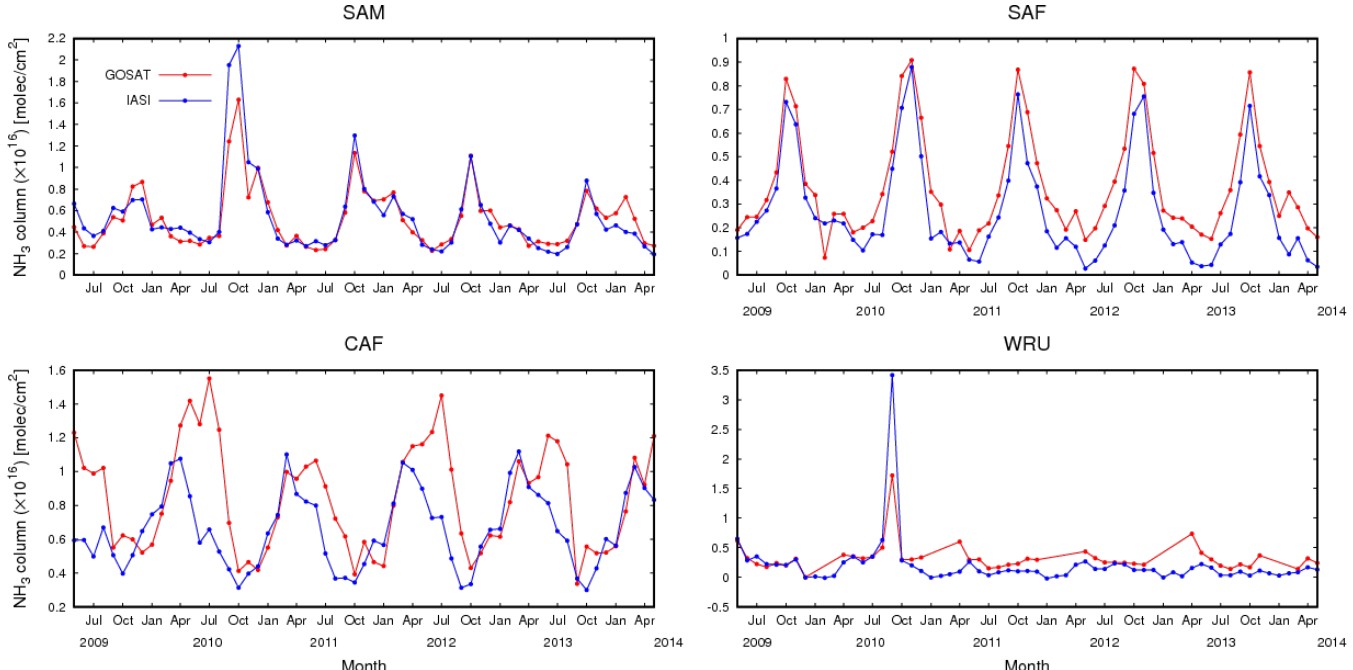

**Figure 10:** Same as Figure 9, but for the biomass burning areas.

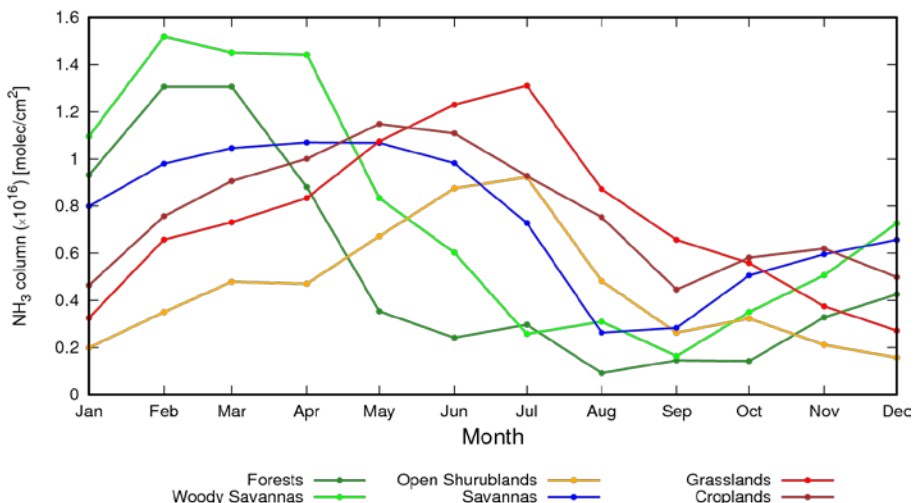

**Figure 11.** Monthly variations of ammonia column amount from the GOSAT for the study period over forests, woody savannas, open shrublands, savannas, grasslands, and croplands classified by the MODIS product in CAF.