# Peer review of "Atmospheric ammonia retrieval from the TANSO-FTS/GOSAT thermal infrared sounder"

_Atmospheric Measurement Techniques, 2019_

## Referee Comment (RC1) · Anonymous Referee #1 · 23 Jul 2019

Adding a GOSAT ammonia product with other already available satellite derived NH3 will be beneficial to the community. This paper demonstrates that with sufficient averaging that GOSAT can capture the general global spatial patterns of ammonia seen by other sensors and emission inventories. The paper also points out some differences between the emission inventories (e.g. over Central Asia) and with IASI. In general the paper is well written, and other authors did a good job at addressing most of the initial comments. There are three main comments, and a number of more minor ones that should be addressed.

Main Comments: 1) One nice result from this paper is the potential impact of dust on the ammonia retrievals. The paper presently states on Page 8: "The presence of dust can lead to the overestimation of ammonia because of the wavenumber dependence

of its absorption properties. Figure 13 shows an example of the observed spectra contaminated by dust aerosols over Saharan desert. The residual shows some similarity of wavenumber dependencies of ammonia signals on the spectra.". The wavelength dependence of dust in the longwave infrared being similar to sharp ammonia spectra is very surprising. In general would expect the optical properties from the dust to have broad spectral features. The authors will need to show the input optical properties of the dust (e.g. emissivity, absorption cross-sections, etc.) and the LBLRTM radiative transfer forward model calculations showing the resulting difference in the spectra with and without the dust (keeping the rest of the input atmospheric state the same).

Why does the dust only impact ammonia retrievals over some deserts around the globe? Does the moisture play a role? Is it the dust composition (e.g. type of sand)? Is there ammonia mixed in with the dust (transport)? These results can be shown in an appendix and just referred to in the paper.

2) Page 3: The authors did add in much more details on the GOSAT sensor capabilities in the revised manuscript, which is a great help in determining GOSAT for ammonia retrievals. However, it would be good to help address the question of why one might want to use GOSAT when there are other sensors with more coverage, such as IASI, CrIS, and AIRS. I would suggest highlighting the higher spectral resolution of GOSAT compared to other sensors more upfront. There is some information provided later in the paper on Page 8 the authors mention a bit about the sensitivity when discussing potential differences between GOSAT and IASI (Page 8 lines 3-7). However, the spectral resolution is only part of the signal-to-noise to determine the sensitivity of the instrument. Thus, to investigate it more fully one needs to determine how much the slightly higher spectral noise of GOSAT mitigates some of the potential sensitivity gains from the higher spectral resolution. It would be great if an estimate of the sensitivity (e.g. minimum detection limit) could computed. Again, this might be done by performing simulations where the truth is known. The forward model calculations (LBLRTM) can be run over a number of atmospheric conditions and varying ammonia amounts. Then the

ammonia spectral signal can be compared with the estimated GOSAT spectral noise (to obtain a signal-to-noise). Under conditions where the signal-to-noise is just above 1 (the NH3 spectral signature can be seen) will indicate the detection limit for GOSAT. One other thing that higher spectral resolution provides is the potential to reduce systematic errors (e.g. interfering species) as there is a greater capability to micro-window around weak spectral features from other species reducing the cross-state errors in the GOSAT NH3 retrievals. It would be good to add more to this discussion to help the community with the question of why they might want to use GOSAT NH3 observations.

3) For a new product it is best to show a little validation (e.g. surface, aircraft, or FTIR). In general most FTIR stations have relatively low concentration amounts, however, there are sites like Bremen, Toronto, Mexico City, Pasadena, that obviously have high concentrations that can be compared with GOSAT. GOSAT might even have some special stare observations for these sites as they are also part of TCCON and NDACC networks.

Minor Comments:

4) Page 1: Line 24: "Atmospheric nitrogen is taken up by animals" ???...reword

5) Page 2: Line 30: Change "(Shephard and Cady-Pereira, 2015)" to "Shephard and Cady-Pereira (2015)"

6) Page 3: Lines 2-4: "Since carbon dioxide (CO2) and methane (CH4) concentrations have been previously derived from both regions (Yoshida et al., 2011, 2013; Saitoh et al., 2009, 2016), it was possible to calculate the concentrations of ammonia within the same footprint." Can more context be provided here as to why CO2 and CH4 needs to be retrieved for NH3. It is not directly for the retrieval of NH3 as it is mostly window region where temperature (from CO2) and water vapour retrievals are retrieved before NH3 (as stated in the paper later on).

7) Page 4: Need a "." After (Van Damme et al., 2017).

8) Page 4: Just a comment as a future improvement: The AFGL NH3 is likely very low apriori for most hot-spot regions. That being said, ammonia is not known well globally so there is not a good "climatological" apriori fields to use. As a future GOSAT retrieval refinement, the authors might want to consider using an apriori selection process (e.g. similar to TES and CrIS).

9) Page 4: When showing averaging kernels it is common to show the rows of the averaging kernels, rather than columns of the averaging kernels. It would be useful to show the 2D averaging kernel plot included in the response to the initial Review 1 questions so that the GOSAT vertical sensitivity can be compared against other sensors such as TES, AIRS, and CrIS. Even if it is just added in as an appendix and referred to in the text relative to other sensors that produce averaging kernels.

10) For ease of comparisons, the GOSAT team might want to consider reporting the NH3 retrieval values in units commonly used by the other satellite retrievals (e.g. level values in ppbv and total column in molecules/cm2).

11) Page 5: Figures 3 and 7. Just a suggestion: Is it possible to change the colour bar slightly to highlight a few more hot-spot regions (e.g. North America and Western Europe) without making the plot look very noisy. Maybe bring the yellow and reds reach down to lower values.

12) Page 5, line 19: averaged errors: excluded when estimated concentrations were negative. . . does this mean they excluded negative concentrations in all of their analysis? As that would high bias the results.

13) Page 5: line 20: change "highconcentrations" to "high concentrations".

14) Page 5: line 21: add in the reference for the TES global plots showing the typical hotspots (Shephard et al., 2011).

15) Page 6, line 5-6: The high concentrations can be explained by soil emissions, Hickman et al., 2018: see https://www.atmos-chem-phys.net/18/16713/2018/acp-18-

16713-2018.pdf.

16) Page 7: Discussion section: Lines 13 & 14 list possible causes for differences, and then the rest of the paragraph goes into more detail for some of them. The authors might consider merging (3) and (4) together as "signal-to-noise (spectral resolution and noise)". Then order the more detail discussion in the rest of the paragraph in the same order as listed so it is so that it is easy to follow.

17) Page 8: See main comment above on dust.

18) Page 17: It would be good to expand the EUR to include Northern Italy and NE Spain as both are known hotspots in Europe.

19) Acknowledgements: I believe there is a user agreement on how to acknowledge the use of the IASI NH3 product. As there are no IASI NH3 product developers as coauthors there should be an acknowledgement stated here for using the IASI data.
* * *

---

## Referee Comment (RC2) · Anonymous Referee #2 · 5 Aug 2019

The manuscript "Atmospheric ammonia retrieval from the TANSOFTS/GOSAT thermal infrared sounder" introduces a recent product of ammonia from GOSAT. This is an important topic and should be published. I do have a few questions that I hope the authors can address before the manuscript is accepted.

Major Comments: 1. The "Discussion" section is relatively weak. I believe the largest difference between GOSAT and IASI is the retrieval/measurement sensitivity due to the thermal contrast differences between the two sensors that take measurements ~3-hours apart. This would explain why the differences (GOSAT-IASI) are smaller or even in opposite sign in the summer when the surface temperatures are the highest. Another evidence for the thermal contrast influence is that the differences between the two sensors are less at low latitudes. Diurnal cycles do contribute to the differences, but

[Figure]

I would think the differences should increase in the summer, not decrease. The other possible causes (1-5) for the ammonia differences are less relevant. For example, "The GOSAT retrieval has a sensitivity in the middle troposphere as mentioned in Sect. 2, and therefore the scaled profile from the GOSAT retrieval likely underestimates the concentrations near the surface in these situations." This is incorrect. Even though the spectral sensitivity is in the mid-troposphere, most of the ammonia concentration is near the surface. Also, if the AFGL profile amount is biased, it would affect the GOSAT retrievals similarly globally, but the large positive differences are at higher latitudes and in colder seasons. Additionally, these differences do not seem to be limited to the agricultural source regions, as discussed. Biomass burning signals are stronger, so all sensors should capture the signals well. 2. There is not enough evidence in its current form to support that the ammonia differences in the central African is due to aerosol contaminations. It could be due to surface emissivity, temperature, etc. Should state instead that the aerosol contamination is a likely cause of the difference in CAF but thorough studies are needed, at a later time. 3. The latter half of the paper is difficult to read. I have the following suggestions: a. In the "Results" section, figures were introduced first, then they are described in the paragraphs. Readers have to go back and force to find the relevant figures to understand the discussion. I suggest adding (see Figure xx) after the main sentences. b. The acronyms, ECH, CUS, EUR, CAF, SAM, WRU, and SEA etc. do not save a lot of space but makes reading much more difficult. I suggest eliminating them, at least in the text. The DJF, MAM, JJA, and SON are fine since they are commonly used. c. Figure 11 and 12 are not very helpful, neither were they discussed thoroughly. I would eliminate them, but this is up to the authors.

Minor Comments: 1. Page 4 Line 29, "The standard deviations of the a priori and the measured spectra for ammonia retrievals were assumed to be 20 and 0.3 K, respectively." What is the unit of 20? 2. Page 9 Line 3, "...which iteratively decreases the difference..." should be which iteratively minimize the difference...

---

## Author Comment (AC1) · 25 Sep 2019

Dear Anonymous Referee #1,

The authors really appreciate for your kind and helpful comments. We respond to the reviewer's comments as follows. The comments from the reviewer are written in blue and the author's response are written in the following. The sentences modified or added in the manuscript are highlighted by red.

Main comments

1) One nice result from this paper is the potential impact of dust on the ammonia retrievals. The paper presently states on Page 8: "The presence of dust can lead to the overestimation of ammonia because of the wavenumber dependence of its absorption properties. Figure 13 shows an example of the observed spectra contaminated by dust aerosols over Saharan desert. The residual shows some similarity of wavenumber dependencies of ammonia signals on the spectra.". The wavelength dependence of dust in the longwave infrared being similar to sharp ammonia spectra is very surprising. In general would expect the optical properties from the dust to have broad spectral features. The authors will need to show the input optical properties of the dust (e.g. emissivity, absorption cross-sections, etc.) and the LBLRTM radiative transfer forward model calculations showing the resulting difference in the spectra with and without the dust (keeping the rest of the input atmospheric state the same). Why does the dust only impact ammonia retrievals over some deserts around the globe? Does the moisture play a role? Is it the dust composition (e.g. type of sand)? Is there ammonia mixed in with the dust (transport)? These results can be shown in an appendix and just referred to in the paper.

We had investigated the dust aerosol impacts on the ammonia absorption band using the radiative transfer code, Pstar (Ota et al. 2010) and LBLRTM. Several refractive indices were obtained from Di Biagio et al. (2014) and Hess et al. (1998) and the simulations were operated using some patterns of the particle size distributions for each. However, we could not reproduce the spectral feature like Fig. 13 in AMTD. As the presence of dust layer can also lead to errors in the temperature and humidity retrievals, the spectral feature may be the results of the combinations of the errors and the impacts from the certain types or compositions of dust aerosols. On the errors and contaminations on the desert area, we have to investigate the impacts more precisely. Therefore, this topic would be a further investigation according to the comment from reviewer#2. The body text was modified and the following sentences were added. In addition, Figure 13 in AMTD was included in Supplement as Fig. S2.

Page 8, line 30: "Figure S2 shows an example of the observed spectra from which a high ammonia value was retrieved in the Sahara. The spectra are V shaped, which are spectral characteristics of dusty conditions. Therefore, a part of the high values in northwestern Africa during JJA were caused by dust contamination. However, further investigation is needed because the composition of dust and its impact are quite complicated. Moreover, other error sources such as surface emissivity and humidity also exist."

2) Page 3: The authors did add in much more details on the GOSAT sensor capabilities in the revised manuscript, which is a great help in determining GOSAT for ammonia retrievals. However, it would be good to help address the question of why one might want to use GOSAT when there are other sensors with more coverage, such as IASI, CrIS, and AIRS. I would suggest highlighting the higher spectral resolution of GOSAT compared to other

sensors more upfront. There is some information provided later in the paper on Page 8 the authors mention a bit about the sensitivity when discussing potential differences between GOSAT and IASI (Page 8 lines 3-7). However, the spectral resolution is only part of the signal-to-noise to determine the sensitivity of the instrument. Thus, to investigate it more fully one needs to determine how much the slightly higher spectral noise of GOSAT mitigates some of the potential sensitivity gains from the higher spectral resolution. It would be great if an estimate of the sensitivity (e.g. minimum detection limit) could computed. Again, this might be done by performing simulations where the truth is known. The forward model calculations (LBLRTM) can be run over a number of atmospheric conditions and varying ammonia amounts. Then the ammonia spectral signal can be compared with the estimated GOSAT spectral noise (to obtain a signal-to-noise). Under conditions where the signal-to-noise is just above 1 (the NH3 spectral signature can be seen) will indicate the detection limit for GOSAT. One other thing that higher spectral resolution provides is the potential to reduce systematic errors (e.g. interfering species) as there is a greater capability to micro-window around weak spectral features from other species reducing the cross-state errors in the GOSAT NH3 retrievals. It would be good to add more to this discussion to help the community with the question of why they might want to use GOSAT NH3 observations.

We assumed that the spectral accuracy of the retrieval is 0.3K. However, in our simulation, the column amount corresponding to the signal of this value in the strongest channel is more than $1.0\times10^{16}$ [molec/cm$^2$] which is much larger in the lower values shown in Figures. Therefore, we guess that the random noise is smaller than that. The following text was added in the manuscript and the figure of the signals for the different spectral resolution was also added in Supplemental as Fig. S3.

Page 8, line 1: "Cause (2) is related to the detection limits of the instruments used. The spectral resolution of GOSAT is higher than that of IASI, AIRS, and CrIS, which have fine coverage. Because fine spectral resolution provides a stronger signal in the spectra, it provides a high signal-to-noise. Moreover, high spectral resolution can reduce the contaminations from interfering species, such as water vapor, by channel selection. The maximum signal is approximately 0.04 K or 0.05 K for a spectral resolution of 0.5 or 0.6, which corresponds to AIRS, IASI, and CrIS. On the other hand, it is approximately 0.1 K for a resolution of 0.2, which corresponds to GOSAT (Fig. S3). This indicates that the signal of ammonia in the GOSAT spectra is approximately twice as strong as those in the other sounders. In Section 2.1, we assumed a spectral noise of 0.3 K. If the relation between the signal and the ammonia concentration is linear, then the ammonia column amount corresponding to this is approximately $1.4 \times 10^{16}$ molec/cm$^2$. This value is considerably higher than the minimum values seen in the previous figures. Based on these figures, the lower limit of GOSAT retrieval seems to be approximately $0.2 \times 10^{16}$–$0.4 \times 10^{16}$ molec/cm$^2$. This corresponds to a value lower than 0.1 K in the spectra. Although our assumption of spectral noise is based on Kataoka et al. (2013), the reported values not only include random noise but also spectral biases. Therefore, our results imply that the random noise of the GOSAT spectral region used in the ammonia retrieval is approximately 0.1 K.
"

3) For a new product it is best to show a little validation (e.g. surface, aircraft, or FTIR). In general most FTIR stations have relatively low concentration amounts, however, there are sites like Bremen, Toronto, Mexico City, Pasadena, that obviously have high concentrations that can be compared with GOSAT. GOSAT might even have

some special stare observations for these sites as they are also part of TCCON and NDACC networks.

We investigate the match-up data between GOSAT and the FTIR at Bremen, Toronto, Mexico City, and Pasadena with the condition according to Dammers et al. (2017). Unfortunately, there are no co-located measurements at Toronto, Mexico City, and Pasadena due to the coarse geometry of the GOSAT observation. Since there were co-located measurements geometrically at Bremen, I requested to Prof. Palm at University of Bremen and obtained the FTIR data. However, there were only five match-up measurements. Therefore, we couldn't find enough data for validations.

Minor comments

4) Page 1: Line 24: "Atmospheric nitrogen is taken up by animals" ???...reword
The sentence was modified as follows.
Page 1, line 24: "Nitrogen is taken up by animals and plants and is emitted into the atmosphere when organic matter decays or is burned."

5) Page 2: Line 30: Change "(Shephard and Cady-Pereira, 2015)" to "Shephard and Cady-Pereira (2015)"
Page 2, line 30: Corrected.

6) Page 3: Lines 2-4: "Since carbon dioxide ($CO_2$) and methane ($CH_4$) concentrations have been previously derived from both regions (Yoshida et al., 2011, 2013; Saitoh et al., 2009, 2016), it was possible to calculate the concentrations of ammonia within the same footprint." Can more context be provided here as to why $CO_2$ and $CH_4$ needs to be retrieved for $NH_3$. It is not directly for the retrieval of $NH_3$ as it is mostly window region where temperature (from $CO_2$) and water vapour retrievals are retrieved before $NH_3$ (as stated in the paper later on).
We didn't mean $CO_2$ and $CH_4$ need to be retrieved before $NH_3$ retrieval, but we can obtain the $CO_2$ and $CH_4$ concentrations in the same footprint for the combinational use of them. The text was modified as follows.
Page 3, line 4: "Carbon dioxide ($CO_2$) and methane ($CH_4$) concentrations have been derived from both spectral regions (Yoshida et al., 2011, 2013; Saitoh et al., 2009, 2016) and provided as products. Therefore, the combinational use of concentrations of ammonia and these products within the same footprint can be useful to study carbon cycles."

7) Page 4: Need a "." After (Van Damme et al., 2017).
Page 4, line 15: Corrected.

8) Page 4: Just a comment as a future improvement: The AFGL $NH_3$ is likely very low apriori for most hot-spot regions. That being said, ammonia is not known well globally so there is not a good "climatological" apriori fields to use. As a future GOSAT retrieval refinement, the authors might want to consider using an apriori selection process (e.g. similar to TES and CrIS).
Thank you for your comment. We would like to modify a priori selection in the ammonia retrieval using GOSAT-2 which is a successor of GOSAT.

9) Page 4: When showing averaging kernels it is common to show the rows of the averaging kernels, rather than columns of the averaging kernels. It would be useful to show the 2D averaging kernel plot included in the response to the initial Review 1 questions so that the GOSAT vertical sensitivity can be compared against other sensors such as TES, AIRS, and CrIS. Even if it is just added in as an appendix and referred to in the text relative to other sensors that produce averaging kernels.

The 2D averaging kernel plot was added as Figure.S1 in the appendix. The following sentence was added.

Page 4, line 31: "The 2D plot of the averaging kernel matrix is also shown in Fig. S1."

10) For ease of comparisons, the GOSAT team might want to consider reporting the NH3 retrieval values in units commonly used by the other satellite retrievals (e.g. level values in ppbv and total column in molecules/cm2).

Figure 3, 4, 7, 8, 9, 10: Corrected.

11) Page 5: Figures 3 and 7. Just a suggestion: Is it possible to change the colour bar slightly to highlight a few more hot-spot regions (e.g. North America and Western Europe) without making the plot look very noisy. Maybe bring the yellow and reds reach down to lower values.

Figure 3 and 7: Rainbow color bar was corrected. In addition, same color bar was applied to Fig. 4 and 6.

12) Page 5, line 19: averaged errors: excluded when estimated concentrations were negative. . . does this mean they excluded negative concentrations in all of their analysis? As that would high bias the results.

No, they are not excluded in the analysis of the horizontal distribution or time series of the concentrations.

13) Page 5: line 20: change "highconcentrations" to "high concentrations".

Corrected.

14) Page 5: line 21: add in the reference for the TES global plots showing the typical hotspots (Shephard et al., 2011).

Page 6, line 1: "TES (Shephard et al., 2011)" was added.

15) Page 6, line 5-6: The high concentrations can be explained by soil emissions, Hickman et al., 2018: see https://www.atmos-chem-phys.net/18/16713/2018/acp-18- C4 AMTD Interactive comment Printer-friendly version Discussion paper 16713-2018.pdf.

The following sentence was added.

Page 8, line 25: "In another study, Hickman et al. (2018) reported that the high concentrations of $NH_3$ in March and April observed by IASI are related to the soil moisture."

16) Page 7: Discussion section: Lines 13 & 14 list possible causes for differences, and then the rest of the paragraph goes into more detail for some of them. The authors might consider merging (3) and (4) together as "signal-to-noise (spectral resolution and noise)". Then order the more detail discussion in the rest of the paragraph in the

same order as listed so it is so that it is easy to follow.

The spectral resolution and noise were merged as (2) signal-to-noise. The discussion section was reconstructed along with the order as listed in the first paragragh.

17) Page 8: See main comment above on dust.

18) Page 17: It would be good to expand the EUR to include Northern Italy and NE Spain as both are known hotspots in Europe.

The area was modified. Figure 5 and Table 2 were also modified.

19) Acknowledgements: I believe there is a user agreement on how to acknowledge the use of the IASI NH3 product. As there are no IASI NH3 product developers as coauthors there should be an acknowledgement stated here for using the IASI data.

Page 10, line3: The following text was added in the acknowledgements.

[revised manuscript text omitted]

**Supplementals**

[Figure]

**Figure S1.** An example of the averaging kernel matrix of the ammonia profile retrieval using the AFGL ammonia profile and mid-latitude summer atmosphere.

[Figure]

**Figure S2.** An example of the observed spectra by the GOSAT affected by dust aerosols at (0.86°E, 29.17°N) on 22 July 2010.

[Figure]

**Figure S3.** Brightness temperature differences between the case assuming and not assuming ammonia for spectral resolutions of 0.2 – 0.6. Mid-latitude summer profile was used and ammonia column amount of $4.56 \times 10^{15}$ molec/cm$^2$ was assumed. The instrumental line shape function was assumed as that of GOSAT.

---

## Author Comment (AC2) · 25 Sep 2019

Dear Anonymous Referee #2,

The authors really appreciate for your kind and helpful comments. We respond to the reviewer's comments as follows. The comments from the reviewer are written in blue and the author's response are written in the following. The sentences modified or added in the manuscript are highlighted by red.

Major comments

1. The "Discussion" section is relatively weak. I believe the largest difference between GOSAT and IASI is the retrieval/measurement sensitivity due to the thermal contrast differences between the two sensors that take measurements ~3-hours apart. This would explain why the differences (GOSAT-IASI) are smaller or even in opposite sign in the summer when the surface temperatures are the highest. Another evidence for the thermal contrast influence is that the differences between the two sensors are less at low latitudes. Diurnal cycles do contribute to the differences, but I would think the differences should increase in the summer, not decrease. The other possible causes (1-5) for the ammonia differences are less relevant. For example, "The GOSAT retrieval has a sensitivity in the middle troposphere as mentioned in Sect. 2, and therefore the scaled profile from the GOSAT retrieval likely underestimates the concentrations near the surface in these situations." This is incorrect. Even though the spectral sensitivity is in the mid-troposphere, most of the ammonia concentration is near the surface. Also, if the AFGL profile amount is biased, it would affect the GOSAT retrievals similarly globally, but the large positive differences are at higher latitudes and in colder seasons. Additionally, these differences do not seem to be limited to the agricultural source regions, as discussed. Biomass burning signals are stronger, so all sensors should capture the signals well.

The text in the Discussion was modified as that the largest cause of the differences is thermal contrast and the following were added or modified.

Page 7, line 24: "The fact that the differences are smaller at high latitudes also supports the large contribution of thermal contrast differences."

Page 7, line 27: "However, the differences caused by diurnal cycles of ammonia should be larger during summer. Therefore, the contribution from this seems to be smaller than that from the thermal contrast."

In Summary, the sentence was modified as follows.

Page 9, line 29: "The main cause of these differences seems to be temporal gaps in the observations, especially caused by the thermal contrast between surface skin and air."

The sentence about the profile shape was modified as follows.

Page 8, line 16: "Cause (4) possibly affects the column integration of ammonia amounts. In this study, the ammonia profile shape is assumed to be that of the AFGL profiles. However, the vertical gradients of the concentrations can be considerably larger in source regions and considerably smaller in clear air. On the other hand, IASI retrieval uses a priori profiles from the model calculations, and these vertical gradients are larger than those of the AFGL profile. Although differences between the profile shapes of GOSAT and IASI are equally present across the globe, the magnitude of relations between the estimated values vary for each area. This suggests that this effect is not so apparent in the other causes."

2. There is not enough evidence in its current form to support that the ammonia differences in the central African is due to aerosol contaminations. It could be due to surface emissivity, temperature, etc. Should state instead that the aerosol contamination is a likely cause of the difference in CAF but thorough studies are needed, at a later time.

Although the possibility of the dust aerosol contaminations are still in the text, the description which note further investigation is needed was added in the manuscript. The following text was added or modified. Figure 13 in AMTD (example of observed spectra in Sahara) is included in Supplementals.

Page 8, line 30: "Figure S2 shows an example of the observed spectra from which a high ammonia value was retrieved in the Sahara. The spectra are V shaped, which are spectral characteristics of dusty conditions. Therefore, a part of the high values in northwestern Africa during JJA were likely caused by dust contamination. However, further investigation is needed because the composition of dust and its impact are quite complicated. Moreover, other error sources such as surface emissivity and humidity also exist."

3. 3. The latter half of the paper is difficult to read. I have the following suggestions:

   a. In the "Results" section, figures were introduced first, then they are described in the paragraphs. Readers have to go back and force to find the relevant figures to understand the discussion. I suggest adding (see Figure xx) after the main sentences

   The section was reconstructed according to the comment. The descriptions of the figures are introduced as the following in the start of Result section.

Page5, line 15: "Figure 3 contains seasonal maps of the horizontal distributions of the ammonia column amounts obtained from GOSAT within $2.5° \times 2.5°$ grids for each season. Considerably high values were observed above several areas such as India, eastern China, and central Africa. Figure 4 shows the averaged errors within each grid. When the estimated amounts were negative, the values were excluded. The errors were approximately $0.5 \times 10^{16}$ molec/cm$^2$. The high concentration areas are highlighted in Fig. 5 and listed in Table 2. Figure 6 depicts the distributions of anthropogenic ammonia emissions obtained from version 2 of the Emission Database for Global Atmospheric Research–Hemispheric Transport of Air Pollution (EDGAR-HTAP; Janssens-Maenhout et al., 2012), and from GFED4.1s. Similar to Fig.3, the ammonia column amounts over land obtained from IASI are shown in Fig. 7. The time period of these observations was the same as that for the GOSAT data and was from 23 April 2009 to 14 December 2014. The data were filtered to obtain data collected only during the day and in the absence of clouds. The differences between Fig. 3 and Fig. 7 are shown in Fig. 8. The time series of the monthly averaged ammonia column amounts obtained from GOSAT and IASI for six agricultural emission areas (central U.S., Europe, central Asia, India, Southeast Asia, and eastern China) and four biomass burning areas (South America, central Africa, South Africa, and western Russia) are outlined in Table 2 and are shown in Fig. 9 and 10, respectively."

   b. The acronyms, ECH, CUS, EUR, CAF, SAM, WRU, and SEA etc. do not save a lot of space but makes reading much more difficult. I suggest eliminating them, at least in the text. The DJF, MAM, JJA, and SON

are fine since they are commonly used
In the body text, the acronyms were restored.

c. Figure 11 and 12 are not very helpful, neither were they discussed thoroughly. I would eliminate them, but this is up to the authors.
Figure 11 and 12 were eliminated in the manuscript.

Minor comments

1. Page 4 Line 29, "The standard deviations of the a priori and the measured spectra for ammonia retrievals were assumed to be 20 and 0.3 K, respectively." What is the unit of 20?
   20 is unitless because it is the value about scaling factor. The text was modified as follows.
   Page 4, line 29: "The standard deviations of the a priori scaling factor and the measured spectra for ammonia retrievals were assumed to be 20 and 0.3 K, respectively."

2. Page 9 Line 3, ". . .which iteratively decreases the difference. . ." should be which iteratively minimize the difference. . .
   The text was modified as follows.

[revised manuscript text omitted]

---

## Editor Decision (ED1)

Editor's Comments on amt-2019-49
**Atmospheric ammonia retrieval from the TANSO-FTS/GOSAT thermal infrared sounder by Yu Someya, Ryoichi Imasu, Kei Shiomi, and Naoko Saitoh**

Thank-you for your detailed responses to the comments of the two referees. I have reviewed them and the revised manuscript and have some follow-up comments below. Please submit a brief response addressing each of these comments and a new version of the manuscript.

The order of comments below corresponds to the order in the response to referees.

Response to Referee #1

Page 8, line 26 and Figure S2: "The spectra are V shaped, which are spectral characteristics of dusty conditions."
- It's not clear what is meant by a "V-shaped spectrum" nor how this is seen in Figure S2. Please use a more informative description.

Page 7, line 29 to page 8, line 9: This entire paragraph needs careful revision, as follows:

"The spectral resolution of GOSAT is higher than that of IASI, AIRS, and CrIS, which have fine coverage.
- It is not clear what is meant by "fine coverage". Clarify. Could restate the values of resolution for each instrument here.

"Because fine spectral resolution provides a stronger signal in the spectra, it provides a high signal-to-noise."
- Is this referring to GOSAT or the other sensors? This is ambiguous given the preceding sentence. Doesn't higher spectral resolution give higher noise and lower SNR? See, e.g., https://www.osapublishing.org/ol/abstract.cfm?uri=ol-39-1-60. Correct this sentence.
- "signal-to-noise" should be changed "signal-to-noise ratio" throughout the manuscript (e.g., page 7, line 8 – do a search and replace).

"Moreover, high spectral resolution can reduce the contaminations from interfering species, such as water vapor, by channel selection."
- Would be clearer to say something like: "by selecting narrower fitting windows that exclude spectral features of other gases."

"The maximum signal is approximately 0.04 K or 0.05 K for a spectral resolution of 0.5 or 0.6, which corresponds to AIRS, IASI, and CrIS. On the other hand, it is approximately 0.1 K for a resolution of 0.2, which corresponds to GOSAT (Fig. S3). This indicates that the signal of ammonia in the GOSAT spectra is approximately twice as strong as those in the other sounders. In Sect. 2.1, we assumed a spectral noise of 0.3 K."
- Add units: "0.5 or 0.6 cm$^{-1}$" and "0.2 cm$^{-1}$".

- Is the "maximum signal" actually the maximum noise, and "the signal of ammonia" the noise? Noise should increase and signal should decrease with spectral resolution, which is the opposite of what is stated in this sentence, although the "spectral noise of 0.3 K" implies that it is the noise that is being described. Is it the GOSAT signal or the GOSAT noise that is twice as large as the other sounders? Please check this text carefully and revise.

"If the relation between the signal and the ammonia concentration is linear, then the ammonia column amount corresponding to this is approximately $1.4 \times 10^{16}$ molec/cm$^2$."
- Briefly explain how this column amount is derived from the spectral noise of 0.3 K.

"This corresponds to a value lower than 0.1 K in the spectra."
- A value of what? Random noise? State this.

Referee #1, comment 3:
- It is unfortunate that there are so few GOSAT TIR coincidences with the NDACC FTIR stations. This is a bit surprising given that these stations have been used to validate GOSAT SWIR and GOSAT TIR measurements (e.g., https://www.atmos-meas-tech.net/10/3697/2017/). Are there fewer GOSAT TIR NH$_3$ measurements than TIR CH$_4$? Page 3, line 5 implies that there are NH$_3$ measurements in the same spatial footprints as CH$_4$. What coincidence criteria were used?
- The referee is correct in that it is preferable to include some validation comparisons for a new product. The term "validation" is not used anywhere in the manuscript although Section 3.2 describes comparisons with IASI. Do these qualify as validation of the GOSAT NH$_3$ product? Could add a sentence noting the need for validation, the lack of coincidences with NDACC FTIR (note time period and coincidence criteria considered), the relevance of the IASI comparisons for this purpose, and scope for comparisons with other satellite measurements of NH$_3$.

Page 3, line 6: "Therefore, the combinational use of concentrations of ammonia and these products within the same footprint can be useful to study carbon cycles."
- Is NH$_3$ part of the carbon cycle? Perhaps rewrite as: "Therefore, the combination of NH$_3$, CO$_2$, and CH$_4$ measurements within the same spatial footprint may be useful for studying linkages between the nitrogen and carbon cycles."

Page 10, line 7: Please check that the correct wording has been used for the acknowledgement of IASI data products.
- For example, this website https://iasi.aeris-data.fr/data-use-policy/, gives specific wording for minor use (e.g., a plot) of IASI NH$_3$:
"IASI is a joint mission of EUMETSAT and the Centre National d'Etudes Spatiales (CNES, France). The authors acknowledge the AERIS data infrastructure for providing access to the IASI data in this study and ULB-LATMOS for the development of the retrieval algorithms."
- However, given that comparisons with IASI are an important part of this manuscript, this may qualify as "substantial use (ie the results would have been different without the IASI dataset). Please contact the principal investigator to offer co-authorship to the team: NH$_3$ Lieven Clarisse lclariss@ulb.ac.be, Pierre-François Coheur pfcoheur@ulb.ac.be". If the authors have not

contacted the IASI team, I strongly recommend doing so to ask whether co-authorship is warranted.

**Response to Referee #2**

Page 8, line 14: "On the other hand, IASI retrieval uses a priori profiles from the model calculations, and these vertical gradients are larger than those of the AFGL profile."
- What model calculations? Provide more information here.
"Although differences between the profile shapes of GOSAT and IASI are equally present across the globe, the magnitude of relations between the estimated values vary for each area."
- Rewrite this sentence to clarify what is meant by "equally present" (implies the same everywhere) and "magnitude of relations".
"This suggests that this effect is not so apparent in the other causes."
- This sentence is ambiguous. What effect? What other causes. Rewrite to be more specific.

Page 9, line 12: "The optimal estimation, which iteratively minimize the difference between the calculated and the observed spectra, was used for analysis."
- Change to: "The optimal estimation method, which iteratively minimizes the difference between the calculated and the observed spectra, was used for analysis."

**Additional Comments**

Page 2, line 19: "Currently, the five space-borne nadir satellite sounders, namely, the Atmospheric Infrared Sounder (AIRS), Tropospheric Emission Spectrometer (TES), Infrared Atmospheric Sounding Interferometer (IASI), Thermal and Near-infrared Spectrometer for Observation-Fourier Transform Spectrometer (TANSO-FTS), and Cross-track Infrared Sounder (CrIS), are available to observe atmospheric ammonia."
- TES ended in January 2018 (https://tes.jpl.nasa.gov/mission/) so this sentence should be revised, e.g., "Five space-borne nadir satellite sounders, namely, the Atmospheric Infrared Sounder (AIRS), Tropospheric Emission Spectrometer (TES), Infrared Atmospheric Sounding Interferometer (IASI), Thermal and Near-infrared Spectrometer for Observation-Fourier Transform Spectrometer (TANSO-FTS), and Cross-track Infrared Sounder (CrIS), have provided observations of atmospheric ammonia that overlap with the GOSAT mission."

Page 3, line 16: Spectral resolution can be defined in multiple ways. I think for GOSAT, it is the full width at half maximum of the instrumental line shape – add this information.

Page 3, line 17: "spectral accuracy" is ambiguous – is this the accuracy of the spectral radiance in units of brightness temperature? State this clearly.

Page 3, line 18: "Kataoka et al. (2013) reported that it is 0.5 K"

Page 3, line 20: "optimal estimation method"

Page 3, lines 20-21: "As noted in Sect. 2.2, we assumed that [what? the accuracy of the spectral radiance in units of brightness temperature? state explicitly] is 0.3 K in the spectral range used in the ammonia retrieval."
Page 3, line 21: "Earth's surface"

Page 7, Section 4: This section is somewhat speculative.  Consider whether the discussion might be tightened up.

Page 7, line 9: "(4) assumed ammonia profile"
- Clarify whether this means the assumed ammonia profile shape, the assumed ammonia a priori profile, or something else.

Page 19, line 6: "data are" (not is, data are plural)

Page 22, line 5: Change caption to "Figure S1. An example of the averaging kernel matrix for the GOSAT TIR ammonia profile retrieval using the AFGL ammonia profile and mid-latitude summer atmosphere."

Page 22, line 8: "An example of a GOSAT TIR spectrum affected by …"

Page 23, line 1: add units for "spectral resolutions of 0.2 – 0.6".  Perhaps rewrite this caption, changing
"Figure S3. Brightness temperature differences between the case assuming and not assuming ammonia for spectral resolutions of 0.2 – 0.6. Mid-latitude summer profile was used and ammonia column amount of $4.56 \times 10^{15}$ molec/cm$^2$ was assumed. The instrumental line shape function was assumed as that of GOSAT."
to
"Figure S3. Brightness temperature differences between spectra simulated with and without including ammonia, using the GOSAT instrumental line shape function with spectral resolutions between 0.2 and 0.6 cm$^{-1}$ as indicated in the legend. These simulations used a mid-latitude summer profile for ammonia with a total column of $4.56 \times 10^{15}$ molec/cm$^2$."

---

## Author Response (AR2)

Dear editor,

Thank you for editing our manuscript and the follow-up comments. We respond to your comments as follows. Editor's comments are described as blue and our responses are in the following. The sentences modified or added in the manuscript are highlighted by red.

Response to Referee#1

Page 8, line 26 and Figure S2: "The spectra are V shaped, which are spectral characteristics of dusty conditions."
- It's not clear what is meant by a "V-shaped spectrum" nor how this is seen in Figure S2. Please use a more informative description.
The sentences were corrected as follows.
Page 8, line 29: "The spectra are V–shaped as a result of decreasing radiances with wavenumbers between 800 cm$^{-1}$ and 1000 cm$^{-1}$ and increasing radiances with wavenumbers between 1060 cm$^{-1}$ and 1240 cm$^{-1}$. This shape is a characteristic of dust contamination due to the spectral dependence of the refractive index."

Page 7, line 29 to page 8, line 9: This entire paragraph needs careful revision, as follows:

"The spectral resolution of GOSAT is higher than that of IASI, AIRS, and CrIS, which have fine coverage.
- It is not clear what is meant by "fine coverage". Clarify. Could restate the values of resolution for each instrument here.
"fine coverage" means "horizontal fine coverage". But it is not necessary here, so it was removed. The sentence was corrected as follows.
Page 7, Line 29: "The spectral resolution of GOSAT (0.265 cm$^{-1}$) is higher than that of IASI (0.5 cm$^{-1}$), AIRS (0.5 cm$^{-1}$), and CrIS (0.625 cm$^{-1}$)."

"Because fine spectral resolution provides a stronger signal in the spectra, it provides a high signal-to-noise."
- Is this referring to GOSAT or the other sensors? This is ambiguous given the preceding sentence. Doesn't higher spectral resolution give higher noise and lower SNR? See, e.g.,
https://www.osapublishing.org/ol/abstract.cfm?uri=ol-39-1-60. Correct this sentence.
- "signal-to-noise" should be changed "signal-to-noise ratio" throughout the manuscript (e.g., page 7, line 8 – do a search and replace).
The sentence was corrected as follows. "signal-to-noise" was corrected to "signal-to-noise ratio" in the manuscript.
Page 7, line 30: "The finer spectral resolution provides better isolation of ammonia signals in the spectra."

"Moreover, high spectral resolution can reduce the contaminations from interfering species, such as water vapor, by channel selection."
- Would be clearer to say something like: "by selecting narrower fitting windows that exclude spectral features of other gases."

The sentence was corrected according to the comment (Page 7, line 32).

"The maximum signal is approximately 0.04 K or 0.05 K for a spectral resolution of 0.5 or 0.6, which corresponds to AIRS, IASI, and CrIS. On the other hand, it is approximately 0.1 K for a resolution of 0.2, which corresponds to GOSAT (Fig. S3). This indicates that the signal of ammonia in the GOSAT spectra is approximately twice as strong as those in the other sounders. In Sect. 2.1, we assumed a spectral noise of 0.3 K."
- Add units: "0.5 or 0.6 cm-1" and "0.2 cm-1".
Corrected (Page 8, line 2, 3).

- Is the "maximum signal" actually the maximum noise, and "the signal of ammonia" the noise? Noise should increase and signal should decrease with spectral resolution, which is the opposite of what is stated in this sentence, although the "spectral noise of 0.3 K" implies that it is the noise that is being described. Is it the GOSAT signal or the GOSAT noise that is twice as large as the other sounders? Please check this text carefully and revise.
The sentences were modified as follows.
Page 8, line 3:" This indicates that the isolation of the ammonia signal in the GOSAT spectra is approximately twice as good as those in the other sounders if they have the same noise levels."

"If the relation between the signal and the ammonia concentration is linear, then the ammonia column amount corresponding to this is approximately $1.4 \times 1016$ molec/cm2."
- Briefly explain how this column amount is derived from the spectral noise of 0.3 K.
The sentence was corrected as follows.
Page 8, line 6: "then the ammonia column amount corresponding to 0.3 K of the ammonia spectral signal at the strongest channel is approximately $1.4 \times 10^{16}$ molec/cm$^2$.

"This corresponds to a value lower than 0.1 K in the spectra."
- A value of what? Random noise? State this.
The sentence was corrected as follows.
Page 8, line 9: "This corresponds to an ammonia spectral signal lower than 0.1 K in the spectra."

Referee #1, comment 3:
- It is unfortunate that there are so few GOSAT TIR coincidences with the NDACC FTIR stations. This is a bit surprising given that these stations have been used to validate GOSAT SWIR and GOSAT TIR measurements (e.g., https://www.atmos-meas-tech.net/10/3697/2017/). Are there fewer GOSAT TIR NH3 measurements than TIR CH4? Page 3, line 5 implies that there are NH3 measurements in the same spatial footprints as CH4. What coincidence criteria were used?
- The referee is correct in that it is preferable to include some validation comparisons for a new product. The term "validation" is not used anywhere in the manuscript although Section 3.2 describes comparisons with IASI. Do these qualify as validation of the GOSAT NH3 product? Could add a sentence noting the need for validation, the lack of

coincidences with NDACC FTIR (note time period and coincidence criteria considered), the relevance of the IASI comparisons for this purpose, and scope for comparisons with other satellite measurements of NH3.

The FTIR data provided by Prof. Palm at Bremen Univ. contains only 240 observations from 2008 to 2014. In Dammers et al. (2017) (https://www.atmos-meas-tech.net/10/2645/2017/amt-10-2645-2017.pdf), the match-up criteria are 90 min and 50 km in maximum. On the other hand, that is 12 h and 500 km in Olsen et al. (2017) (https://www.atmos-meas-tech.net/10/3697/2017/). This difference is the cause of small number of the match-up data for ammonia validation. Because lifetime of ammonia is much shorter than that of methane, the match-up criteria for the methane product is not applicable to the ammonia product.

We don't recognize the comparison with IASI as a validation. The following paragraph was added in Sect. 5.

Page 10, line 3: "Although the results were evaluated by comparing them with the IASI product, they should be validated with the other measurements. The ground-based measurement is appropriate for validation. The IASI and CrIS ammonia products were validated with the ground-based Fourier Transform Infrared Spectroscopy (FTIR) measurement at the Network for the Detection for Stratospheric Change (NDACC) sites (Dammers et al., 2016; Dammers et al., 2017). Unfortunately, very few coincident measurements exist between GOSAT and NDACC FTIR with the match-up criteria shown in the papers (90 min and 50 km) because of the sparse scan geometry of GOSAT. Therefore, we must validate our results as a part of future work. Moreover, other satellite ammonia products exist from AIRS and CrIS. Inter-comparisons with these products are also needed. This should lead to a higher reliability of the satellite products and a deeper understanding of ammonia behavior.

Page 3, line 6: "Therefore, the combinational use of concentrations of ammonia and these products within the same footprint can be useful to study carbon cycles."
- Is NH3 part of the carbon cycle? Perhaps rewrite as: "Therefore, the combination of NH3, CO2, and CH4 measurements within the same spatial footprint may be useful for studying linkages between the nitrogen and carbon cycles."
The sentence was replaced (Page 3, line 6).

Page 10, line 7: Please check that the correct wording has been used for the acknowledgement of IASI data products.
- For example, this website https://iasi.aeris-data.fr/data-use-policy/, gives specific wording for minor use (e.g., a plot) of IASI NH3:
"IASI is a joint mission of EUMETSAT and the Centre National d'Etudes Spatiales (CNES, France). The authors acknowledge the AERIS data infrastructure for providing access to the IASI data in this study and ULB-LATMOS for the development of the retrieval algorithms."
- However, given that comparisons with IASI are an important part of this manuscript, this may qualify as "substantial use (ie the results would have been different without the IASI dataset). Please contact the principal investigator to offer co-authorship to the team: NH3 Lieven Clarisse lclariss@ulb.ac.be, Pierre-François Coheur pfcoheur@ulb.ac.be". If the authors have not contacted the IASI team, I strongly recommend doing so to ask

whether co-authorship is warranted.

The acknowledgement was modified according to the data use policy (Page 10, line 21). We sent the manuscript to Lieven Clarisse and Pierre-François Coheur, and asked the co-authorship. They said that they need not to be added the co-authors. Therefore, they were not added the co-authors.

**Response to Referee #2**

Page 8, line 14: "On the other hand, IASI retrieval uses a priori profiles from the model calculations, and these vertical gradients are larger than those of the AFGL profile."

- What model calculations? Provide more information here.

"Although differences between the profile shapes of GOSAT and IASI are equally present across the globe, the magnitude of relations between the estimated values vary for each area."

- Rewrite this sentence to clarify what is meant by "equally present" (implies the same everywhere) and "magnitude of relations".

"This suggests that this effect is not so apparent in the other causes."

- This sentence is ambiguous. What effect? What other causes. Rewrite to be more specific.

The sentence was modified as follows. According to your comment below, the sentences of speculations were removed.

Page 8, line 15:" On the other hand, IASI retrieval uses fitting function profiles based on the Goddard Earth Observing System chemical transport model and the parameters characterizing the shape of ammonia profile are retrieved (Whitburn et al., 2016)."

Page 9, line 12: "The optimal estimation, which iteratively minimize the difference between the calculated and the observed spectra, was used for analysis."

- Change to: "The optimal estimation method, which iteratively minimizes the difference between the calculated and the observed spectra, was used for analysis."

Corrected (Page 9, line 17).

**Additional Comments**

Page 2, line 19: "Currently, the five space-borne nadir satellite sounders, namely, the Atmospheric Infrared Sounder (AIRS), Tropospheric Emission Spectrometer (TES), Infrared Atmospheric Sounding Interferometer (IASI), Thermal and Near-infrared Spectrometer for Observation-Fourier Transform Spectrometer (TANSO-FTS), and Cross-track Infrared Sounder (CrIS), are available to observe atmospheric ammonia."

- TES ended in January 2018 (https://tes.jpl.nasa.gov/mission/) so this sentence should be revised, e.g., "Five space-borne nadir satellite sounders, namely, the Atmospheric Infrared Sounder (AIRS), Tropospheric Emission Spectrometer (TES), Infrared Atmospheric Sounding Interferometer (IASI), Thermal and Near-infrared Spectrometer for Observation-Fourier Transform Spectrometer (TANSO-FTS), and Cross-track Infrared Sounder (CrIS), have

provided observations of atmospheric ammonia that overlap with the GOSAT mission."

The sentence is corrected as follows.

Page 2, line 18: There are five recent space-borne nadir satellite TIR sounders, namely, the Atmospheric Infrared Sounder (AIRS), Tropospheric Emission Spectrometer (TES), Infrared Atmospheric Sounding Interferometer (IASI), Thermal and Near-infrared Spectrometer for Observation-Fourier Transform Spectrometer (TANSO-FTS), and Cross-track Infrared Sounder (CrIS). AIRS, IASI, TANSO-FTS, and CrIS are still being operated.

Page 3, line 16: Spectral resolution can be defined in multiple ways. I think for GOSAT, it is the full width at half maximum of the instrumental line shape – add this information.

The sentence was corrected as follows.

Page 3, line 16: "The spectral sampling interval is 0.2 cm$^{-1}$ and the full width at half maximum of the instrument line shape is 0.265 cm$^{-1}$."

Page 3, line 17: "spectral accuracy" is ambiguous – is this the accuracy of the spectral radiance in units of brightness temperature? State this clearly.

Page 3, line 18: "Kataoka et al. (2013) reported that it is 0.5 K"

The sentences were modified as follows.

Page 3, line 17: "The accuracy of the spectral radiance of the TIR band of TANSO-FTS has a wavenumber dependency. Kataoka et al. (2013) reported that is it 0.5 K in unit of brightness temperature at 800 – 900 cm$^{-1}$ and 0.1 K at 980 – 1080 cm$^{-1}$."

Page 3, line 20: "optimal estimation method"

Corrected (Page 4, line 20)

Page 3, lines 20-21: "As noted in Sect. 2.2, we assumed that [what? the accuracy of the spectral radiance in units of brightness temperature? state explicitly] is 0.3 K in the spectral range used in the ammonia retrieval."

The sentence was modified as follows.

Page 3, line 19: "We assumed that the accuracy of spectral radiance is 0.3 K in the spectral range used in the ammonia retrieval."

Page 3, line 21: "Earth's surface"

Corrected (Page 3, line 22).

Page 7, Section 4: This section is somewhat speculative. Consider whether the discussion might be tightened up.

As you commented, there are many speculations in section 4. Unfortunately, there are no papers about comparisons of the satellite ammonia products and we couldn't show robust foundations of them. Alternatively, we add the sentence at the end of section.

Page 9, line 13: "The discussions above are only speculations. We must evaluate the observed differences considering other independent observations."

Moreover, the parts of thermal contrast and dust contaminations are not removable in order to address the reviewer's comments. Therefore, we could remove only the speculation about (4) as following. If there are other removable part, point out please.

Page 7, line 9: "(4) assumed ammonia profile"
- Clarify whether this means the assumed ammonia profile shape, the assumed ammonia a priori profile, or something else.

The sentence was modified as follows.

Page 7, line 9: "(4) assumed ammonia profile shape"

Page 19, line 6: "data are" (not is, data are plural)

Page 20, line 6: Corrected.

Page 22, line 5: Change caption to "Figure S1. An example of the averaging kernel matrix for the GOSAT TIR ammonia profile retrieval using the AFGL ammonia profile and mid-latitude summer atmosphere."

Figure S1: Corrected.

Page 22, line 8: "An example of a GOSAT TIR spectrum affected by …"

Figure S2: Corrected.

Page 23, line 1: add units for "spectral resolutions of 0.2 – 0.6". Perhaps rewrite this caption, changing
"Figure S3. Brightness temperature differences between the case assuming and not assuming ammonia for spectral resolutions of 0.2 – 0.6. Mid-latitude summer profile was used and ammonia column amount of $4.56 \times 1015$ molec/cm2 was assumed. The instrumental line shape function was assumed as that of GOSAT."
to
"Figure S3. Brightness temperature differences between spectra simulated with and without including ammonia, using the GOSAT instrumental line shape function with spectral resolutions between 0.2 and 0.6 cm-1 as indicated in the legend. These simulations used a mid-latitude summer profile for ammonia with a total column of $4.56 \times 1015$ molec/cm2."

Figure S3: The caption was changed to the suggested one.

[revised manuscript text omitted]

**Supplementals**

[Figure]

**Figure S1.** An example of the averaging kernel matrix for the GOSAT TIR ammonia profile retrieval using the AFGL ammonia profile
and mid-latitude summer atmosphere.

[Figure]

**Figure S2.** An example of a GOSAT TIR spectrum affected by dust aerosols at (0.86°E, 29.17°N) on 22 July 2010.

[Figure]

**Figure S3.** Brightness temperature differences between spectra simulated with and without including ammonia, using the GOSAT instrumental line shape function with spectral resolutions between 0.2 and 0.6 cm$^{-1}$ as indicated in the legend. These simulations used a mid-latitude summer profile for ammonia with a total column of $4.56 \times 10^{15}$ molec/cm$^2$.